# Microbiota-driven interleukin-17-producing cells and eosinophils synergize to accelerate multiple myeloma progression

Arianna Calcinotto[1,12], Arianna Brevi[1,2], Marta Chesi [3], Roberto Ferrarese[4], Laura Garcia Perez[5], Matteo Grioni[1], Shaji Kumar [6], Victoria M. Garbitt[3], Meaghen E. Sharik[3], Kimberly J. Henderson[6], Giovanni Tonon[7], Michio Tomura[8], Yoshihiro Miwa[9], Enric Esplugues[10], Richard A. Flavell [10], Samuel Huber[5], Filippo Canducci [4,11], Vincent S. Rajkumar[6], P. Leif Bergsagel [3] & Matteo Bellone [1]

The gut microbiota has been causally linked to cancer, yet how intestinal microbes influence progression of extramucosal tumors is poorly understood. Here we provide evidence implying that *Prevotella heparinolytica* promotes the differentiation of Th17 cells colonizing the gut and migrating to the bone marrow (BM) of transgenic Vk*MYC mice, where they favor progression of multiple myeloma (MM). Lack of IL-17 in Vk*MYC mice, or disturbance of their microbiome delayed MM appearance. Similarly, in smoldering MM patients, higher levels of BM IL-17 predicted faster disease progression. IL-17 induced STAT3 phosphorylation in murine plasma cells, and activated eosinophils. Treatment of Vk*MYC mice with antibodies blocking IL-17, IL-17RA, and IL-5 reduced BM accumulation of Th17 cells and eosinophils and delayed disease progression. Thus, in Vk*MYC mice, commensal bacteria appear to unleash a paracrine signaling network between adaptive and innate immunity that accelerates progression to MM, and can be targeted by already available therapies.

[1] Division of Immunology, Transplantation and Infectious Diseases, IRCCS San Raffaele Scientific Institute, Milan 20132, Italy. [2] Vita-Salute San Raffaele University, 20132 Milan, Italy. [3] Comprehensive Cancer Center, Mayo Clinic Arizona, Scottsdale, AZ 85259, USA. [4] Laboratory of Microbiology, IRCCS San Raffaele Scientific Institute, Milan 20132, Italy. [5] Molekulare Immunologie und Gastroenterologie, Universitätsklinikum Hamburg–Eppendorf, Hamburg 20246, Germany. [6] Division of Hematology, Mayo Clinic Rochester, Rochester, MN 55905, USA. [7] Division of Molecular Oncology, IRCCS San Raffaele Scientific Institute, Milan 20132, Italy. [8] Faculty of Pharmacy, Osaka Ohtani University, Osaka 584-8540, Japan. [9] University of Tsukuba, Tsukuba 305-8575, Japan. [10] Department of Immunobiology, School of Medicine, and Howard Hughes Medical Institute Yale University, New Haven 06520, USA. [11] Department of Biotechnology and Life Sciences, University of Insubria, Varese 21100, Italy. [12] Present address: Institute of Oncology Research, Oncology Institute of Southern Switzerland, Bellinzona, Switzerland. These authors contributed equally: Arianna Calcinotto, Arianna Brevi. Correspondence and requests for materials should be addressed to M.B. (email: bellone.matteo@hsr.it)

While many factors regulating cancer progression are tumor cell autonomous, they are insufficient to induce progression to malignancy. Among the cell-extrinsic drivers of cancer, a strong link has been proposed between diet, commensal bacteria, and aerodigestive tract malignancies[1]. Microbes within the gut also contribute to carcinogenesis at mucosal sites by altering the balance of epithelial cell proliferation and death, by favoring the production of toxic metabolites from host-produced factors and drugs, and by promoting chronic inflammation and/or local immune suppression[1].

As the microbiome of each organ is distinct, the effects on inflammation and carcinogenesis are likely to be organ specific[2]. Nevertheless, gut commensal bacteria are involved in the pathogenesis of extramucosal autoimmune diseases[3], thus supporting the role of the gut microbiota in shaping systemic immune responses. Yet, the mechanisms by which non-pathogenic microbes drive non-aerodigestive tract malignancies remain to be elucidated.

Commensal bacteria are involved in the differentiation of Th17 cells[4], which mainly produce IL-17A (also defined IL-17), IL-17F, and IL-22, all cytokines playing a critical role in inflammation[5]. The role of Th17 cells in cancer is controversial. While some authors showed that Th17 cells were efficient in eliminating tumors[6], others reported accumulation of Th17 cells in several tumors, in which they promoted tumor initiation[7] and progression[8].

In multiple myeloma (MM), a B cell neoplasm characterized by the accumulation of clonal plasma cells within the bone marrow (BM), and in most cases a monoclonal protein (i.e., M-spike) in blood and/or urine, Th17 cells have been linked to advanced disease with bone lesions[9]. Of relevance, IL-17 can promote tumor growth through an IL-6-STAT3 signaling pathway[10], which is also pivotal for plasma cell growth[11], thus suggesting a role for IL-17 in different phases of MM.

No data are available on the potential role of IL-17-producing cells in the early, asymptomatic phases of MM, and on the mechanisms by which IL-17-producing cells are induced and/or recruited in the BM of MM patients. Smoldering multiple myeloma (SMM) is an asymptomatic phase that may anticipate full-blown MM. The definition of SMM has been proposed to fill the gray zone between monoclonal gammopathy of undetermined significance (MGUS), a rather common finding in the elders, and active MM. Indeed, patients affected by SMM are subjected to more frequent follow-up than MGUS because they have a much higher risk of progression[12]. However, likely because of heterogeneity in the pathobiology of the disease and lack of adequate risk stratification, few interventional studies in SMM patients have shown improved overall survival with therapy[13]. Indeed, most of the accepted clinical parameters to define high-risk SMM are evidence-based[13]. This paradigm would benefit from a shift that focuses more on the early modifications in the cellular and molecular composition of the BM microenvironment, thus to identify biological culprits of aggressiveness.

We selected MM as a prototypic extramucosal cancer, and investigated here the potential link between gut microbiota, IL-17 and the progression from asymptomatic SMM to active MM.

## Results

***P. heparinolytica* favors MM progression**. To investigate the link between intestinal microbes and extraintestinal cancers, Vk*MYC mice, which develop a de novo disease mimicking MM[14], were housed in animal facilities located in USA (US) and Italy (IT), and monitored within the years 2012–2018 for disease appearance and the presence of M-spike by serum protein electrophoresis[14]. While a monoclonal M-spike was readily detectable by

20 weeks of age in the blood of about 30% Vk*MYC mice housed in US1 and monitored before 2014, age-matched and sex-matched Vk*MYC mice from US2 (monitored after 2015) or IT (monitored between 2012 and 2018) did not show signs of disease for another 10–15 weeks, a time at which more than 60% of the Vk*MYC mice from the US1 colony had a detectable M-spike (Fig. 1a). Irrespective of the animal facility of origin, age-matched and sex-matched wild type (WT) mice, as expected[15], developed M-spikes much later than Vk*MYC (Fig. 1a), and never developed MM (see Fig. 2b). These findings suggested that the environment, and the microbiota in particular, has a pathogenic impact only on those mice whose plasma cells carry driver genetic alterations like the MYC activation, and is not sufficient *per se* to generate the disease in otherwise healthy mice with spontaneous monoclonal gammopathy.

To identify constituents of the microbiota, stools simultaneously collected from mice housed in the different animal facilities before 2014 and after 2015 were subjected to 16S rDNA-based amplicon sequencing. We did not observe statistically significant differences between US1, US2, and IT samples in terms of intra-sample observed species (α-diversity) by Shannon or CHAO1 indexes (Supplementary Fig. 1). Unweighted UniFrac principal component analyses (β-diversity) clearly segregated the three cohorts of mice, and showed large differences in bacterial species between US1 and US2 or IT mice, irrespective of being Vk*MYC or WT (Fig. 1b). Main diversities were found within 8 taxa (Fig. 1c), belonging to the two major phyla hosted in most mammals: Gram negative *Bacteroidetes* and Gram positive *Firmicutes*[16]. More in details, *Bacteroidetes* (*Bacteroidaceae, Prevoltellaceae, Rikenellaceae,* and *S24–7*) were more represented in the feces of US1 animals (approximately 80% in US1, 56% in US2 and 65% in IT), while US2 and IT animals were more colonized by Firmicutes (*Clostridiales;* 1.62 ± 1.3% in US1; 9.46 ± 8.7% in US2, and 7.6 ± 4% in IT). Health reports confirmed the absence of relevant pathogens in the animal facilities, US1 and US2 mice were housed at the same institution, and changes in the microbiota were not apparently related to the diet, because US1 and US2 were fed the same diet (i.e., PicoLab® Rodent Diet 20 5053; LabDiet), whose content of nutrients, minerals, vitamins and calories was comparable to the diet of IT mice (Teklad Global 18% Protein Rodent Diet; Harlan). The documented changes in the microbiota were likely due instead to the different breeding strategies adopted in US1 and US2 (Fig. 1b). Whereas Vk*MYC breeding in US1 was made by crossing Vk*MYC with WT littermates, the US2 colony was generated by breeding Vk*MYC with C57BL/6J mice from the Jackson Lab. Thus, the microbiota imported from the purchased mice might have modified the microbiota in the US2 colony.

We sought a direct causative role of the microbiota in favoring MM development by treating IT WT mice with a combination of different wide-spectrum antibiotics (ciprofloxacin and metronidazole), and by leaving a group untreated. To perform a controlled study on genetically homogeneous tumors, antibiotic treated and untreated mice were challenged with Vk*MYC-derived Vk12598 cells, a reliable MM model (i.e., t-Vk*MYC MM; see refs.[17,18]). Antibiotic treatment was prolonged for the entire duration of the experiment, and mice were followed for M-spike appearance (Supplementary Fig. 2a). As expected[17], three weeks after transplantation the paraprotein was measurable in sera of 80% of untreated mice, but none of the mice treated with antibiotics showed signs of disease (Fig. 1d and Supplementary Fig. 2b). This did not appear to be due to a direct effect of the antibiotics on plasma cell survival, because the M-spike appeared later in several antibiotic-treated mice (Fig. 1d). Importantly, at the time that all the untreated t-Vk*MYC MM mice with M-spike succumbed of the disease, all antibiotic-treated mice were still

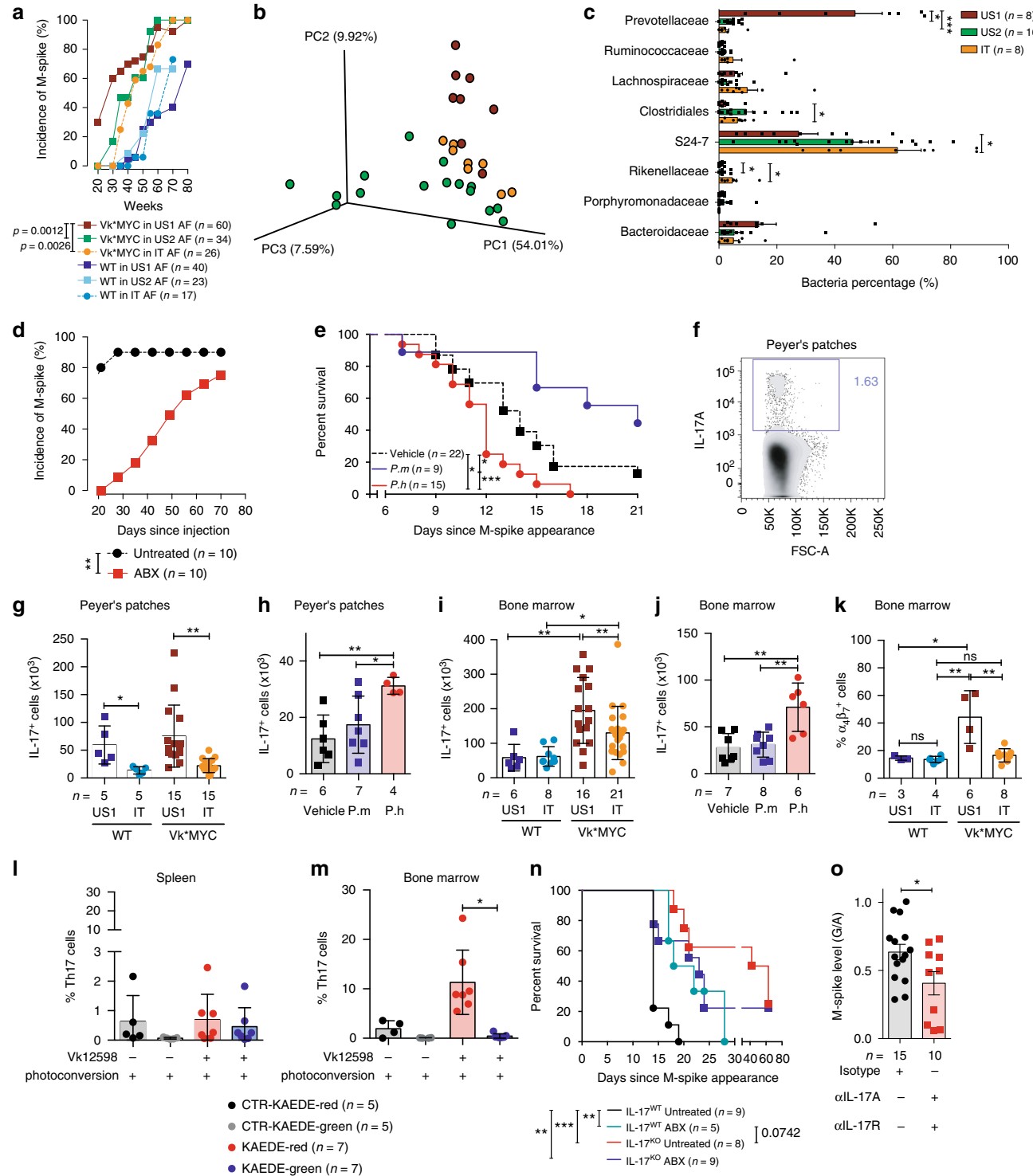

alive, and overall survival was improved in the latter group (Supplementary Fig. 2b).

To further support the link between gut microbiota and MM progression, antibiotic-treated IT mice housed in an isolator were subjected to gavage administration of *Prevotella heparinolytica*, the *Prevotellaceae* mostly represented in US1 (Fig. 1c), before challenge with Vk12598 cells. As control, IT mice were infected with *P. melaninogenica*, which has been associated in humans with improved glucose metabolism under high-fiber diet[19], and in humanized mice with aggressive type II collagen induced arthritis[20]. While in t-Vk*MYC MM mice infected with

*P. heparinolytica*, disease was accelerated, as demonstrated by reduced animal survival when compared to mock-gavaged mice, infection with *P. melaninogenica* prolonged animal survival (Fig. 1e). Thus, in these experimental conditions, microbiota constituents, and *P. heparinolytica* in particular, favor the generation of a microenvironment prone to tumor cell engraftment and expansion.

**P. heparinolytica favors induction of IL-17-producing cells.** A causative link has been proposed between gut microbiota, chronic inflammation mediated by IL-17-producing cells and cancer[21–23].

**Fig. 1** *P. heparinolytica* favors MM progression by promoting BM accrual of IL-17⁺ cells. **a** M-spike incidence over time (weeks) in sex-matched Vk*MYC and WT littermates housed in US1, US2 and IT animal facilities (AF) as indicated in the text. Statistical analyses (Two-way ANOVA). **b** Principal component analysis of fecal microbiota from mice housed in the indicated shelters. **c** Mean ± SD of eight taxa in fecal microbiota between US1 ($n = 8$ of biologically independent mice, US2 ($n = 16$) and IT ($n = 8$) mice relative to total number of reads recovered from each group. Statistical analyses (One-way ANOVA (Dunn's multiple comparison test)): *$P < 0.05$, **$P < 0.01$, ***$P < 0.001$. **d** M-spike incidence over time (days) in t-Vk*MYC MM mice either maintained or not under antibiotics. Unpaired $t$ test: Vehicle vs ABX up to 40 days: *$P < 0.05$. **e** Overall survival (Kaplan–Meier plot) of t-Vk*MYC MM mice gavaged with vehicle (Vehicle), *P. heparinolytica* (P.h) or *P. melaninogenica* (P.m). log-rank (Mantel–Cox) test: Vehicle vs P.h: $P = 0.0157$; Vehicle vs P. m: $P = 0.0346$, P.m vs P.h: $P = 0.0002$. **f** Representative dot plot of Peyer's Patches IL-17⁺ cells (gated on live cells). **g–j** Number of Peyer's Patches (**g, h**) and BM IL-17⁺ cells (**i, j**) from mice described in **a** (**g, i**) and **e** (**h, j**), respectively. **k** Quantification of $\alpha_4\beta_7{}^+$ cells gated on IL-17⁺ cells from the same samples shown in **i**. Mean ± SD of three independent experiments. Unpaired $t$ test: *$P < 0.05$; **$P < 0.01$. **l–m** Frequency of KAEDE red positive (black and red columns) and negative (gray and green columns) Th17 cells in the spleen (**l**) and BM (**m**) of photoconverted control and diseased Kaede mice. Mean ± SD of three independent experiments. Wilcoxon matched-pairs signed rank test; *$P = 0.0156$. **n** Survival (Kaplan–Meier plot) of t-Vk*MYC MM mice IL-17 competent (IL-17ᵂᵀ) or deficient (IL-17ᴷᴼ) and maintained or not maintained under antibiotics. log-rank (Mantel–Cox) test: *$P = 0.0332$, **$P = 0.0021$. **o** M-spike levels are expressed as total gamma globulins/albumin ratio (G/A) in mice within the indicated cohort. Unpaired $t$-test: *$P < 0.05$. **a–e, g–o** $n$, number of mice used

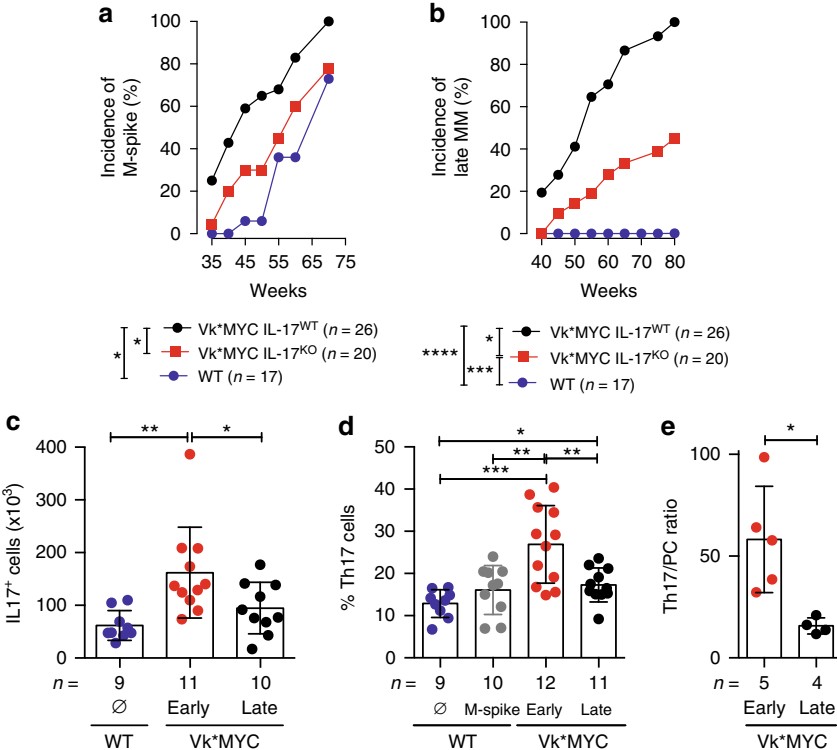

**Fig. 2** Pro-tumoral role of IL-17 during the early phase of MM. **a** M-spike incidence over time (weeks) in cohorts of Vk*MYC mice either competent (Vk*MYC IL-17ᵂᵀ) or deficient for IL-17 (Vk*MYC IL-17ᴷᴼ) and WT littermates. Unpaired $t$ test: *$P < 0.05$. **b** Incidence of M-spike ≥ 6%, corresponding to symptomatic, Late-MM[33], in the mice depicted in **a**. Unpaired $t$ test: *$P < 0.05$; **$P < 0.01$; ***$P < 0.001$; ****$P < 0.0001$. **c, d** Absolute numbers (**c**) and frequency (**d**) of IL-17⁺ cells in the BM of Vk*MYC mice and age-and sex-matched WT littermates. Each dot is representative of an individual mouse. Mean ± SD of five independent experiments. Unpaired $t$ test: *$P < 0.05$; **$P < 0.01$; ***$P < 0.001$. **e** Ratio between Th17 cells and malignant plasma cells (IRF4/MUM1⁺). Mean ± SD of five independent experiments. Whitney test: *$P < 0.0159$. **c–e** Specific $n$ values of biologically independent mice are shown

Interestingly, *Prevotellaceae*, which were almost only present in US1 animals (Fig. 1c), were included among the strains able to promote Th17 differentiation locally and at distant sites[24]. Thus, we searched for IL-17-producing cells in the small intestine of mice housed in the different conditions. A population of IL-17⁺ cells (Fig. 1f) was clearly detectable by flow cytometry analysis in the Peyer's patches of all examined mice (Fig. 1g and Supplementary Fig. 3a, c) in the absence of overt signs of gut inflammation. The number and frequency of IL-17⁺ cells was higher in US1 than in IT mice, and was not influenced by the disease (Fig. 1g and Supplementary Fig. 3a, c), thus confirming that the microbiota of US1 mice and not the pathogenic background of Vk*MYC mice favored the local expansion of IL-17-producing

cells. Also administration of *P. heparinolytica* but not of *P. melaninogenica* induced expansion of IL-17⁺ cells in the gut of t-Vk*MYC MM mice housed in the isolator (Fig. 1h).

To find a correlation between gut microbiota and MM, we looked for IL-17-producing cells in the BM, which is the primary site of MM in both humans and Vk*MYC mice[11,14], of Vk*MYC mice housed in the different conditions. IL-17⁺ cells were enriched in the BM of Vk*MYC mice when compared to WT mice housed in the respective facilities (Fig. 1i and Supplementary Fig. 3b, d). Accumulation of IL-17⁺ cells was more pronounced in the BM of US1 versus IT Vk*MYC mice, whereas, no difference in the number and frequency of these cells was detected in the BM of WT mice housed in either facility

(Fig. 1i and Supplementary Fig. 3b, d). IL-17+ cells were also enriched in the BM of t-Vk*MYC MM receiving *P. heparinolytica* but not of *P. melaninogenica* (Fig. 1j). Thus, a pathologic substrate is required in the BM for IL-17+ cell accumulation, which is favored by selected bacteria.

A direct link between gut microbiota and enrichment of IL-17-producing cells in the BM in Vk*MYC mice was suggested by the presence of a significant proportion of IL-17+ cells expressing the gut-homing integrin α4β7[25] in the BM of Vk*MYC housed in US1 (Fig. 1k). To prove the migration of IL-17+ cells from the gut into the BM of mice affected by MM, we took advantage of the photoconvertible protein Kaede, which permanently changes fluorescence emission from green to red upon photoactivation, and backcrossed Kaede-transgenic mice[26], into IL-17A FP635 reporter knock in mice[27], to generate Kaede/IL-17 mice. Age- and sex-matched Kaede/IL-17 littermates were either sham-treated or injected with Vk12598 cells, and monitored for disease progression. At the appearance of M-spike, the small intestine of all Kaede/IL-17 littermates was photoconverted[28], and the animals were euthanized 60 h later to investigate migration of IL-17+ cells. Whereas the frequency of both Kaede red+ (i.e., IL-17+ cells migrated from the gut) and Kaede green+ (i.e., non-photoconverted IL-17+ cells) cells within CD3+CD4+ cells (Supplementary Fig. 4) was similar in the spleen of both control and Vk12598-challenged mice (Fig. 1l), Kaede red+ cells substantially increased in the BM of tumor-bearing mice (Fig. 1m), thus demonstrating that the presence of MM induced migration of IL-17+ cells from the gut to the BM.

These correlative findings prompted us to look for a causative role of microbiota-driven IL-17 in MM pathogenesis. Thus, Vk12598 cells were either injected in age- and sex-matched IL-17WT or IL-17KO littermates[29] and treated or not with antibiotics. Disease was substantially delayed in IL-17KO mice when compared to WT animals (Fig. 1n). As expected (Fig. 1d), antibiotic treatment prolonged survival of tumor bearing IL-17WT mice (Fig. 1n), but had no effects on IL-17KO mice (Fig. 1n), indicating that IL-17 links the microbiota to MM. These findings also confirmed that antibiotics did not have direct effects on plasma cell survival. To further support our conclusion, Vk12598-challenged mice were treated with either isotype control antibodies or anti-IL17A and anti-IL17R antibodies. Treatment with anti-IL17A and anti-IL17R antibodies delayed MM development (Fig. 1o). Altogether, these findings support a direct link between microbiota diversity, and expansion of a gut-born population of IL-17+ cells that also preferentially accumulate in the BM of mice with MM, and contribute to the pathogenesis of the disease.

**IL-17 accelerates progression of asymptomatic MM.** As our data, together with previous in vitro and in vivo results with human samples[9,30–32], suggested a role for IL-17 in favoring MM aggressiveness, we backcrossed Vk*MYC mice into IL-17KO congenic mice, and monitored them for disease occurrence. Appearance of de novo disease was significantly delayed in Vk*MYC IL-17KO mice when compared with Vk*MYC IL-17WT littermates (Fig. 2a). Additionally, disease progression (i.e., M-spike ≥ 6%, which is characteristic of symptomatic, Late-MM; see ref. [33]) was delayed in Vk*MYC IL-17KO mice (Fig. 2b) when compared to Vk*MYC IL-17WT mice, thus demonstrating that IL-17 is also a precocious propeller of MM in this model. As expected, WT mice never progressed to MM (Fig. 2b).

As our results suggested that IL-17 is involved in early phases of disease (Fig. 2a), we quantified IL-17+ cells (Fig. 2c) in the BM of both asymptomatic (Early)- and symptomatic Late-MM Vk*MYC mice[33]. Surprisingly, a more significant accumulation

of IL-17+ cells was evident in the early phases of MM than in Late-MM (Fig. 2c).

Several immune cells produce IL-17[5]. Indeed, the BM of Vk*MYC mice contained measurable populations of CD3+CD4+ (Supplementary Fig. 5a), CD11b+Gr1+ (Supplementary Fig. 5b), Nk1.1+ CD90.2+ (Supplementary Fig. 5c) and Lin−CD90+CD127+ cells producing IL-17 (Supplementary Fig. 5d), of which T helper type 17 (Th17) cells were the most represented (Supplementary Fig. 5e). Again, a more significant accumulation of Th17 cells (Fig. 2d), and a higher ratio between Th17 cells and neoplastic plasma cells were present in the early phases of MM (Fig. 2e), thus supporting the concept that IL-17-producing cells exert a relevant pathogenic role during the asymptomatic phase and promote MM progression. BM accumulation of Th17 cells was not a characteristic of all aged mice, or a peculiarity of mice with M-spike, because WT mice, either with or without M-spike, did not show enrichment of Th17 cells (Fig. 2d).

Having found accumulation of Th17 cells in the BM of Vk*MYC mice in the early phases of MM, we sought to investigate if such milieu favored Th17 differentiation. Thus, naïve CD4+ T cells from TCR transgenic OTII mice[34] were cultured in the presence of BM serum from either sex-matched and age-matched WT or Vk*MYC mice affected by Early-MM or Late-MM. As control, naïve CD4+ T cells were cultured in the presence of IL-6, TGF-β1, anti-IL-4, and anti-IFN-γ at concentrations known to induce Th17 polarization[35]. Th17 cells were mostly induced by the BM sera from Vk*MYC mice (Fig. 3a), thus confirming that the BM becomes an ideal microenvironment for Th17 cells during disease development in Vk*MYC mice. All together, our findings suggested that IL-17 has a peculiar role in the early phases of disease in Vk*MYC mice.

To mechanistically explain the role of IL-17-producing cells in MM, we assessed the presence of IL-17R in Vk*MYC plasma cells by flow cytometry. As reported in human MM plasma cells[31], Vk*MYC plasma cells (Supplementary Fig. 6) also expressed both subunits of the IL-17R (Fig. 3b), which was functional because exposure to saturating amounts of recombinant IL-17 induced STAT3 phosphorylation in Vk*MYC plasma cells, similarly to saturating amounts of IL-6 (Fig. 3c, e). Interestingly, IL-17 contained in the BM sera from Vk*MYC mice induced STAT3 phosphorylation, and the addition of anti-IL17 antibodies inhibited this phenomenon (Fig. 3d, e). Thus, the BM milieu of Early-MM Vk*MYC mice is rich in soluble factors favoring a Th17 switch, and sustaining neoplastic plasma cells.

Because of transgene expression, all Early-MM Vk*MYC mice are bound to develop symptomatic MM[14]. In contrast, only a fraction of patients with SMM progresses to MM[36], although their plasma cells also express IL-17R (Fig. 4a). Hypothesizing that disease progression in Vk*MYC mice faithfully recapitulates MM progression in SMM patients, we retrospectively measured at SMM diagnosis IL-17 levels in the BM of a cohort of patients that rapidly progressed to MM (i.e., <3 years), and compared these data with those obtained from a cohort of SMM patients that did not progress to MM in the same time frame (Supplementary Table 1). Already at the diagnosis, SMM patients progressing to MM within three years had much higher BM IL-17 than patients not progressing to MM within the same time frame (Fig. 4b). IL-17 did not further increase in MM patients either at diagnosis or after treatment (Fig. 4b). Thus, the content of IL-17 in the BM sera of SMM patients appears to be predictive of progression to symptomatic disease.

**IL-17 activates eosinophils in the BM of Vk*MYC mice.** BM sera of SMM patients were investigated for the content of additional inflammatory chemokines and cytokines. While IL-17 was

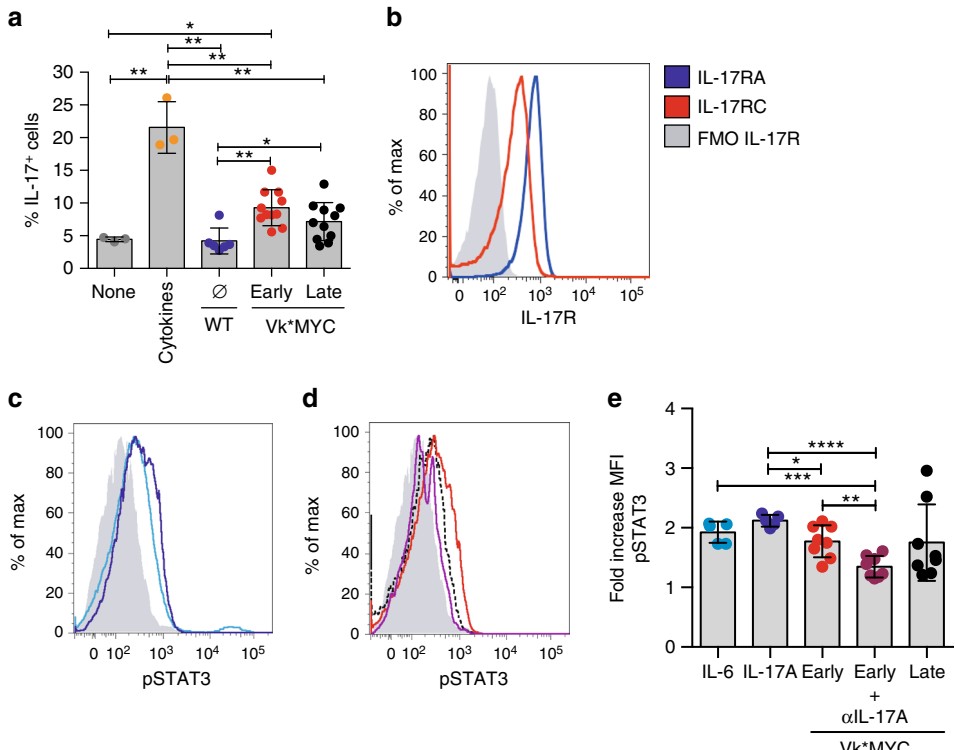

**Fig. 3** IL-17 promotes STAT-3 phosphorylation in Vk*MYC plasma cells. **a** Th17 polarization of OT-II splenocytes cultured for 7 days with BM serum obtained from WT, Early-MM and Late-MM Vk*MYC mice, and assessed for intracellular cytokine release by flow cytometry. None and Cytokines refer to the culture condition with or without IL-6, TGF-β1, anti-IL-4, and anti-IFN-γ antibodies, respectively. (None n = 3, Cytokine n = 3, WT n = 6, Vk*MYC Early n = 11, Vk*MYC Late n = 11). Mean ± SD of three independent experiments. Unpaired t test: *P < 0.05; **P < 0.01; ***P < 0.001. **b** Plasma cells were also stained with anti-IL-17RA and anti-IL-17RC antibodies (blue and red line respectively) and analyzed by flow-cytometry; FMO (Fluorescence Minus One) sample was not stained for IL-17R (gray histogram). **c**, **d** Representative histograms and **e** quantification of Vk*MYC PCs cultured in the presence of either one of the following stimuli: saturating amounts of IL6 (light blue line) or IL-17 (dark blue line), or BM sera from Early- (red line) or Late-MM (black dotted line), or BM sera from Early-MM and anti-IL17 antibodies (purple line). After culture, plasma cells were analyzed by flow-cytometry for STAT3 phosphorylation (pSTAT3). (IL-6 n = 5, IL-17A n = 5, Vk*MYC Early n = 8, Vk*MYC Early + αIL-17A n = 8, Vk*MYC Late n = 8). Mean ± SD of triplicate independent determinations. Unpaired t test: *P < 0.05; **P < 0.01; ***P < 0.001; ****P < 0.0001

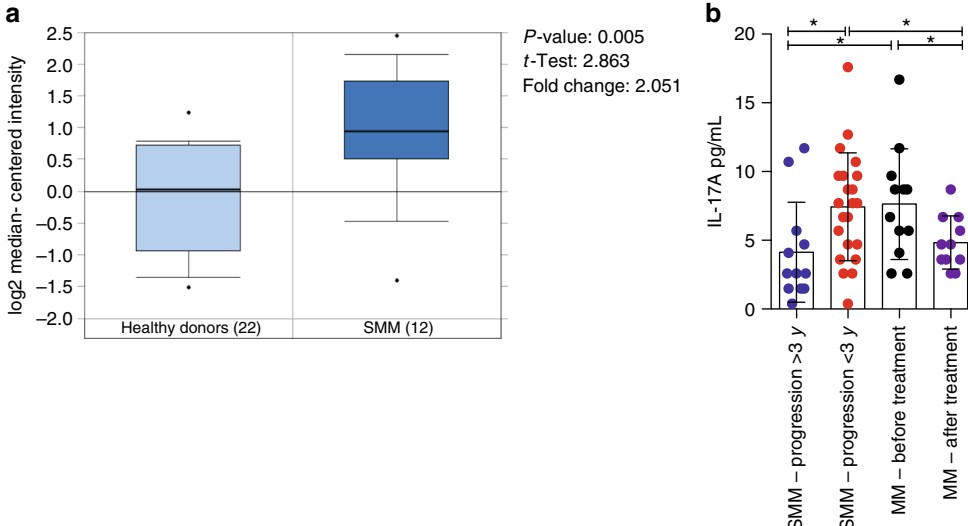

**Fig. 4** IL-17 levels are increased in the BM of SMM patients rapidly progressing to MM. **a** mRNA expression of IL-17RA in primary SMM cells of a cohort of 12 newly-diagnosed patients and 22 matched controls (bone marrow) described in ref.[65]. The expression pattern for the probe set 205707_at is shown. Statistical analysis (Student t test) is reported. **b** IL-17 levels in the BM plasma of SMM patients that progressed to MM within 3 years since the diagnosis (i.e., <3 years), or did not progress to MM in the same time frame (i.e., >3 years). Each dot represents an individual patient. (SMM-Progression > 3 n = 12, SMM-Progression < 3 n = 22, MM-Before treatment n = 12, MM-After treatment n = 11). Data are reported as mean ± SD. Unpaired t test: *P < 0.05

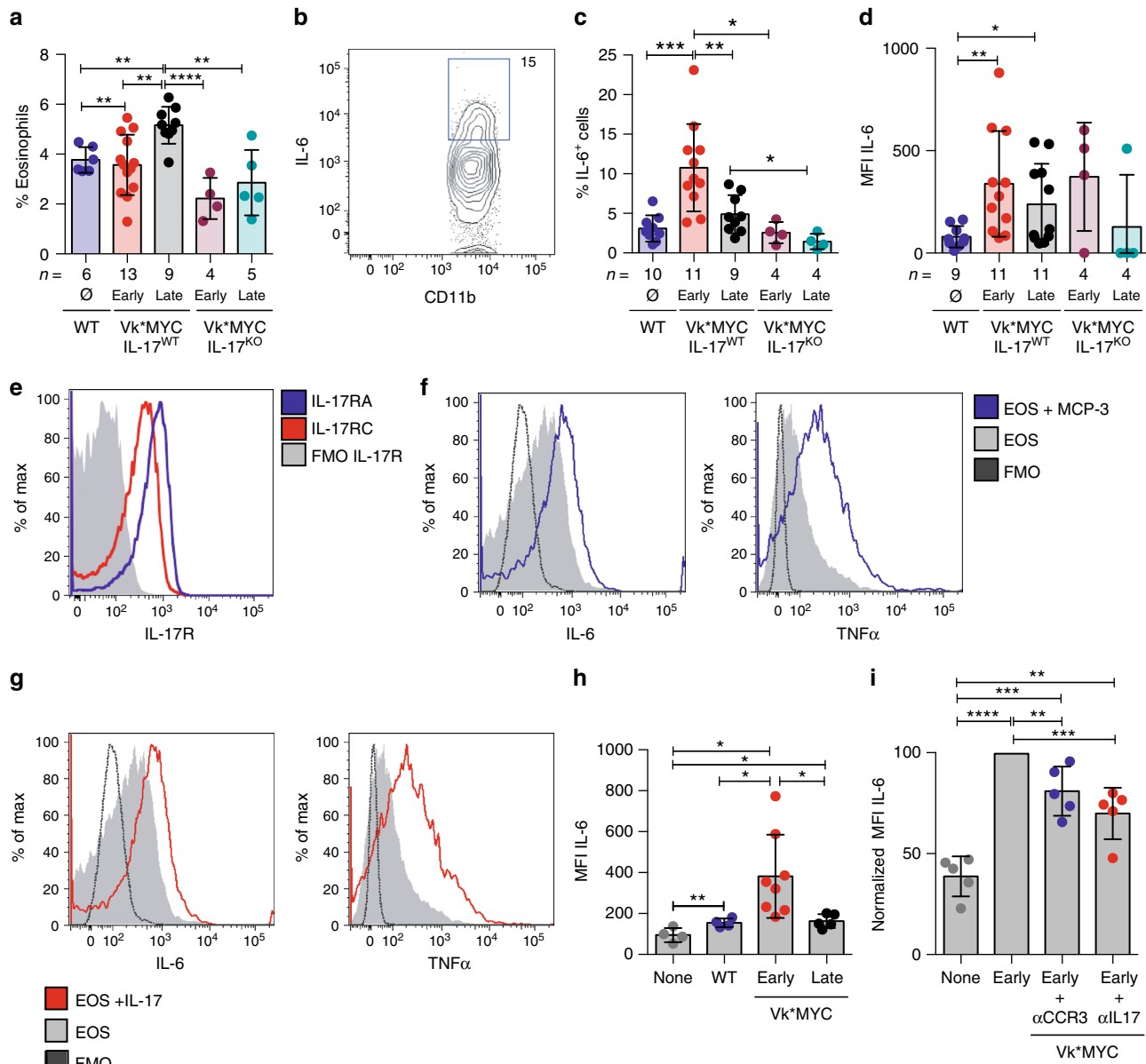

**Fig. 5** An IL-17-eosinophil axis in the BM of Vk*MYC mice favors disease progression. **a** Frequency of BM eosinophils (i.e., CD11b+Ly6C$^{int}$MHC-II$^-$Ly6G$^-$SSC$^{hi}$ or CD11b+Siglec-F+ cells) in Vk*MYC IL-17$^{WT}$ and Vk*MYC IL-17$^{KO}$ mice and age- and sex-matched WT littermates. Each dot represents an individual mouse. Mean ± SD of five independent experiments. Unpaired $t$ test: *$P < 0.05$; **$P < 0.01$; ***$P < 0.001$; ****$P < 0.0001$. **b** Representative dot plot of IL-6+ cells (gated on CD11b+Siglec-F+ cells) in the BM. **c** Percentage of IL-6+ cells gated on CD11b+Siglec-F+ cells. Mean ± SD of five independent experiments. Unpaired $t$ test. **d** Mean fluorescence intensity (MFI) of IL-6 within Siglec-F+CD11b+ cells. Mean ± SD of five independent experiments. Unpaired $t$ test. **e** BM derived eosinophils were also stained with anti-IL-17RA and anti-IL-17RC antibodies (blue and red line respectively) and analyzed by flow-cytometry; FMO (Fluorescence Minus One) sample was not stained for IL-17R (gray histogram). **f** Representative histograms of IL-6 and TNF-α production by eosinophils after IL-17A stimulation (red line). FMO samples were not stained for IL-6 or TNF-α. **g** Representative histograms of IL-6 and TNF-α production by eosinophils after MCP-3 stimulation (blue line). FMO samples were not stained for IL-6 or TNF-α. **h** IL-6 levels (MFI normalized on FMO sample) in eosinophils cultured alone (None; $n = 4$), or in the presence of WT ($n = 4$) or Early-MM ($n = 8$) or Late-MM ($n = 5$) BM serum. Mean ± SD of aggregated data from three independent experiments. Unpaired $t$ test. **i** IL-6 levels (MFI normalized on Early-MM sample) in eosinophils cultured alone (None; $n = 5$), or in the presence of Early-MM with or without the addition of anti-CCR3 ($n = 5$) or anti-IL-17A ($n = 5$). Mean ± SD of aggregated data from three independent experiments. Paired $t$ test. **a**, **c**, **d** Specific $n$ values of biologically independent mice are shown

the only one significantly increased, several other inflammatory factors attracting and activating eosinophils (i.e., RANTES, IFN-γ, IL-4, IL-13, GM-CSF, and IL-5) showed a trend toward enrichment in the BM sera of SMM patients rapidly progressing to MM (Supplementary Fig. 7). Eosinophils play crucial roles both in plasma cell homing to the BM and their retention in the BM niche[37]. Herein, they specifically co-localize with plasma cells[38], and release the proliferation inducing ligand APRIL and

IL-6, essential survival factors for long-lived plasma cells[37]. Eosinophils were indeed present in the BM of Vk*MYC IL-17$^{WT}$ mice developing de novo MM, and their frequency increased with disease progression (Fig. 5a and Supplementary Fig. 8a). Interestingly, eosinophils were not increased in the BM of MM Vk*MYC IL-17$^{KO}$ mice (Fig. 5a). When these cells were assessed for cytokine production, which is a marker of activation, increased frequency of IL-6+ eosinophils (Fig. 5b and

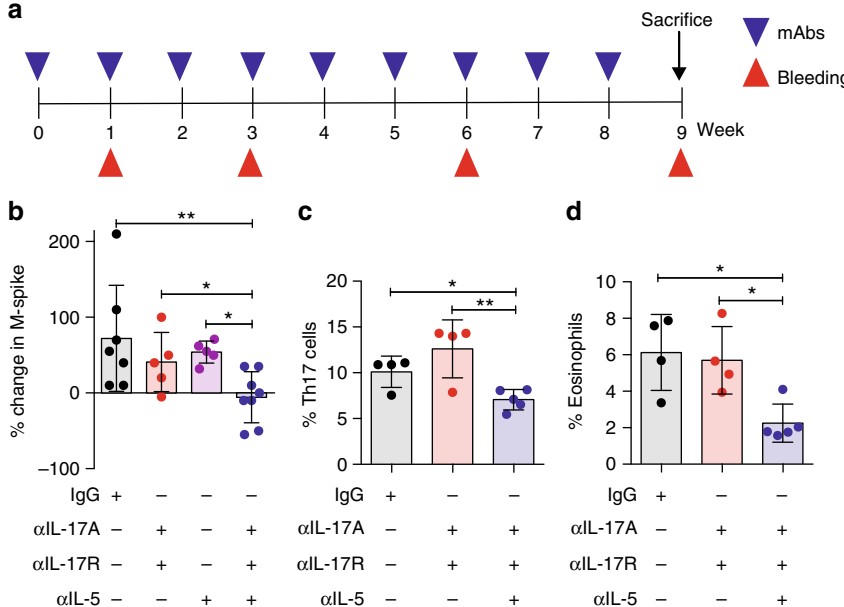

**Fig. 6** IL-17-eosinophil axis neutralization delays disease progression in Vk*MYC mice. **a** Schematic representation of the experiment. **b** Percentage change in M-spike in mice within the indicated cohort (Isotype and αIL-17A, αIL-17R, αIL-5: $n = 8$ mice/group, αIL-17A, αIL-17R and αIL-5: $n = 5$ mice/group) during the observation period. Frequency of BM Th17 (i.e., CD3$^+$CD4$^+$IL-17$^+$) cells **c** and eosinophils (i.e., CD11b$^+$Ly6C$^{int}$MHC-II$^-$Ly6G$^-$SSC$^{hi}$ **d**) was assessed by flow cytometry. Each dot represents an individual mouse. **c** Mean ± SD of two independent experiment. (Isotype $n = 4$, αIL-17A, αIL-17R $n = 4$, αIL-17A, αIL-17R, αIL-5 $n = 5$). Unpaired $t$ test: *$P < 0.05$; **$P < 0.01$. **d** Mean ± SD of two independent experiment. (Isotype $n = 4$, αIL-17A, αIL-17R $n = 4$, αIL-17A, αIL-17R, αIL-5 $n = 5$). One-way ANOVA $P = 0.0101$

Supplementary Fig. 8b) were found in the BM of Early-MM but not Late-MM Vk*MYC IL-17$^{WT}$ mice (Fig. 5c). Again, the lack of IL-17 prevented eosinophil accumulation in the BM of Vk*MYC IL-17$^{KO}$ mice affected by MM (Fig. 5c). Finally, the eosinophil mean fluorescence intensity (MFI) for IL-6 was also increased in Early-MM Vk*MYC mice (Fig. 5d), thus suggesting that eosinophil activating factors were enriched in the BM of these mice, particularly at early phases of disease. Consistently, we detected a trend toward higher levels of MCP-3, which attracts and activates eosinophils[39], in the BM of Early-MM mice when compared to Late-MM (Early-MM 376.9 ± 128.5 pg/mL (mean ± SE; $n = 7$); Late-MM 100.2 ± 15.45 pg/mL ($n = 5$); WT 169.8 ± 46.7 ($n = 5$); Supplementary Fig. 9).

As it has been reported that cytokines of the IL-17 family induce human eosinophils to release cytokines[40], we investigated if IL-17 induced murine eosinophils to produce cytokines. Thus, BM precursors from WT mice were induced in vitro to differentiate to eosinophils (Supplementary Fig. 8c, d; see ref. [41]), and checked for IL-17 receptor expression. In vitro differentiated eosinophils expressed both IL-17RA and IL-17RC subunits (Fig. 5e), and upon stimulation with MCP-3 or IL-17, produced both IL-6 and TNF-α (Fig. 5f, g, respectively). Finally, we investigated if MCP-3 and/or IL-17 levels in the BM of Early-MM mice were sufficient to activate eosinophils. Indeed, eosinophils produced more IL-6 when cultured in the presence of BM serum from Early-MM than Late-MM or WT mice (Fig. 5h). This phenomenon was due to the presence of MCP-3 and IL-17 because the addition of either blocking anti-CCR3 or anti-IL-17 antibodies substantially reduced IL-6 production (Fig. 5i).

**IL-17-eosinophil axis neutralization delays MM progression.**
To determine whether breaking the immune axis between IL-17 and eosinophils delayed disease progression, Early-MM Vk*MYC mice were treated with a cocktail of monoclonal antibodies directed against IL-17RA, IL-17A and IL-5 (Fig. 6a), the latter

being relevant for activation, recruitment and survival of eosinophils[42]. Indeed, treatment with anti-IL-5 antibodies has been shown to reduce eosinophil numbers in blood and BM of mice[43]. Primary end point of the study was to demonstrate that the M-spike in treated mice did not reach values > 6%, as Vk*MYC mice with M-spike ≥ 6% are in the symptomatic MM phase[33]. The combination of the 3 monoclonal antibodies significantly delayed disease progression, which associated with reduced accumulation of both Th17 cells (Fig. 6c) and eosinophils (Fig. 6d) in the BM of Vk*MYC mice. Interestingly, the combination of anti-IL17RA and anti-IL-17A, did not significantly impact the disease (Fig. 6b), and treatment with only anti-IL5 antibodies, while associated with reduced BM accrual of eosinophils (Supplementary Fig. 10c, d), neither impacted disease progression in our MM models (Fig. 6b and Supplementary Fig. 10a), nor affected Th17 accrual in the BM of t-Vk*MYC MM mice (Supplementary Fig. 10b). All together, these data support the concept that disease progression in Vk*MYC mice is propelled by the IL-17-eosinophil axis, which can be broken by the combination of cytokine-specific antibodies (Fig. 7).

**Discussion**
Mammals have co-evolved with their surrounding microbial environment into a complex super-organism, of which commensalism and mutualism are the most advantageous relationships. Conversely, altered host-microbiota interactions drive mucosal inflammation, autoimmunity and aerodigestive tract malignancies[1]. Our findings substantially extend this evidence, demonstrating that *P. heparinolytica*, a commensal bacterium, has a marked effect on the aggressiveness of extramucosal tumors, and independently of gut inflammation. Indeed, we provide evidence that accumulation within the BM of IL-17 producing cells, a phenomenon propelled by a commensal microbe in the absence of overt signs of gut inflammation, is a tumor cell-extrinsic mechanism driving progression of MM, and possibly other extramucosal malignancies.

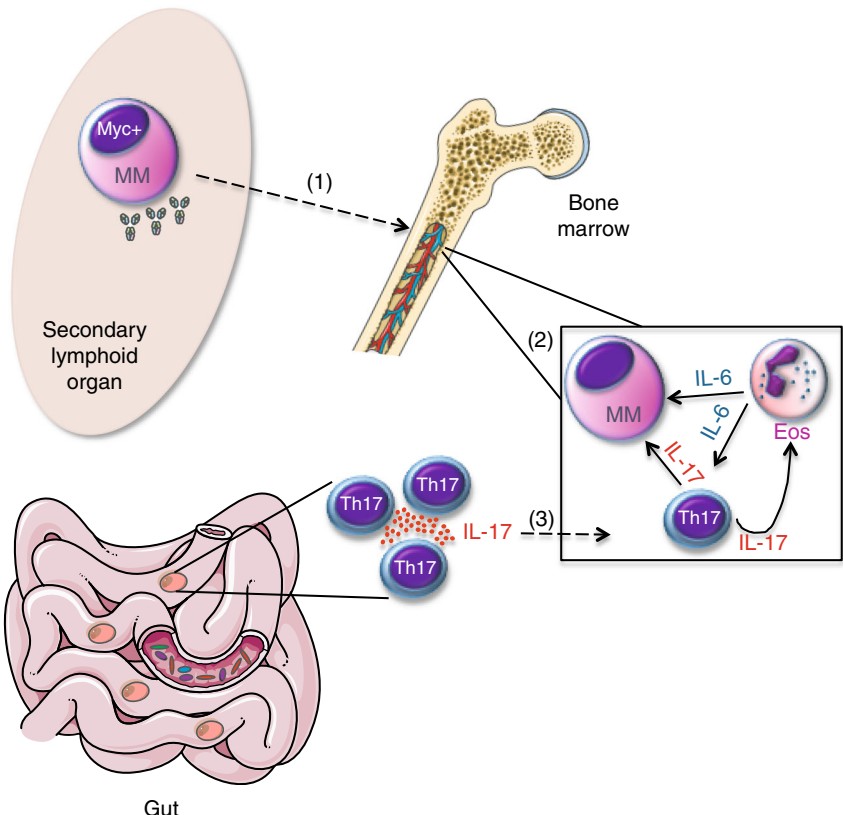

**Fig. 7** IL-17-producing cells indeuced by gut microbiota favor MM progression. (1) Upon AID-dependent MYC activation in germinal centers, a B cell stochastically acquires the characteristics of malignant plasma cell (MM) and migrates to the BM. (2) Within the BM niche, a favorable cytokine milieu induces Th17 skew and eosinophil (Eos) activation, thus establishing a positive-feedback loop that is self-amplifying, and sustains MM progression. (3) A selected gut microbiota locally favors the expansion of Th17 cells, which migrate to the BM niche, where they further contribute to the eosinophil-Th17-MM cells network

Our data also support the existence of a direct immunological link between the gut and the BM, and, more importantly, between the gut and the progression from asymptomatic to symptomatic MM. Thus, we provide mechanistic insights into what has been proposed by Enzeler and colleagues[44], who showed that antimicrobial therapy prevented solid tumor development in partially immunodeficient mice. While a substantial amount of data support a direct link between gut microbiota and gastrointestinal cancer[1], less is known on the potential role of intestinal microbes in extramucosal tumors[45]. As an example, a correlation has been clearly found between orogastric infection with the pathogen *H. hepaticus* and mammary carcinoma[46], through a mechanism that requires innate immunity[47]. Others have elegantly linked TLR5-signaling, microbiota, innate immunity, and extramucosal tumors[23]. Thus, our data extend these previous findings showing that in fully immunocompetent mice, non-pathogenic commensal microbes expand a population of IL-17 producing cells, able to migrate to the BM, and to support MM progression.

*Prevotellaceae*, which are known to promote Th17 differentiation locally and at distant sites[24], were almost only present in US1 animals, and *P. heparinolytica* accelerated MM progression. Thus, *Prevotella* species are primary suspects also in humans, in which the increased abundance of these bacteria at mucosal sites has been associated with Th17-mediated diseases including periodontitis[24] and rheumatoid arthritis[48]. Interestingly, in the humanized HLA-DQ8 murine model, treatment with *P. histicola* but not *P. melaninogenica* suppressed collagen-induced rheumatoid arthritis[20], and *P. histicola* suppressed experimental autoimmune encephalomyelitis by modulating IL-17 production[49]. On this line, an increased abundance of *Prevotella* species

has been associated with reduced intestinal Th17 cell frequency and high disease activity in multiple sclerosis[50]. All together, these findings suggest that selected members of the same genus have different disease modulating properties in different diseases[51].

At the metagenomic level, the microbiota is rather redundant[52], and different classes of bacteria but through similar pathways may drive cancer-promoting effects. Thus, we favor the hypothesis that alterations in microbial richness and function rather than true dysbiosis may affect extramucosal carcinogenesis[2], likely through the fine tuning of the immune response. Accordingly, at pathologic examination, we did not find relevant signs of inflammation in the gut of US1 animals. Rather, expansion of IL-17+ cells in US1 mice might be more likely driven by a different proportion of autobiont species, which are permanent members of the normal commensal microbiota[16]. Unfortunately, loss of *Prevotellaceae* and additional potentially relevant strains in US2 colony did not allow performing adequate experiments of fecal microbiota transplantation to conclusively demonstrate this hypothesis. We speculate that the gut microbiota might also impact human MM. Therefore, identification of the microbiome of selected groups of cancer patients, and altering the composition of the gut microbiota, could be beneficial not only in the prevention of gastrointestinal cancer, but also in delaying progression to symptomatic MM. It will be interesting to investigate these issues by fecal microbiota transplantation or selected bacteria infections in germ-free animals developing MM.

Mechanistically, we have identified the BM milieu of Vk*MYC mice as a microenvironment rich in factors favoring eosinophils and T cells to produce cytokines promoting neoplastic plasma cell survival and expansion. It has been previously reported that IL-17

is systemically rather than locally upregulated in TLR5-unresponsive tumor-bearing mice, but only accelerates malignant progression in IL-6-unresponsive tumors[23]. Our findings challenge this notion, and support a promoting role for gut-driven IL-17 also in IL-6-dependent MM. As MM is an IL-6-driven neoplasm, it would be expected that in patients with TLR5[R392X] polymorphism[23], increased IL-17 serum levels would not favor MM progression. Thus, it would be interesting to verify if in slowly progressing SMM patients TLR5[R392X] polymorphism correlates with high serum levels of IL-17.

Commensal microbes are not unique in favoring the expansion of pathogenic Th17 cells in MM. As an example, mineral oil, which is used in food, cosmetics and biomedicine, has been reported to promote plasma cells neoplasms in BALB/c mice[53], through IL-6[54], eosinophils[43], and possibly the expansion of Th17 cells[55]. Thus, we speculate that several environmental factors in addition to the gut microbiota substantially influence MM progression by inducing pathogenic Th17 cells.

Prior studies have shown that Th17 cells are increased and serum levels of IL-17 are elevated in the BM of symptomatic MM patients[9,30,31], and contribute to myeloma pathology by sustaining malignant plasma cell proliferation and osteoclastogenesis[9,31,32]. Several novel pieces of experimental evidence contained herein extend the role of Th17 cells to the asymptomatic phase of MM. Firstly, M-spike appearance was substantially delayed in Vk*MYC IL-17[KO] mice, thus suggesting that IL-17 is needed for the correct generation of the plasma cell niche within the BM. Additionally, the frequency of Th17 cells and the ratio between Th17 cells and plasma cells in the BM were the highest in Early-MM Vk*MYC mice, which nicely correlated with increased levels of IL-17 in the BM of SMM patients that more rapidly progressed to MM. Together with the recent evidence that Th17 cells are also enriched in the BM of some MGUS-SMM patients[56], our data support a much earlier role for IL-17 in this neoplasm.

The pro-tumor activity of IL-17 is not limited to its direct effect on neoplastic plasma cells expressing the IL-17R, but also through the local activation of eosinophils. Indeed, eosinophils were induced to produce TNF-α, IL-6 and likely others tumor-promoting cytokines upon stimulation with IL-17. IL-6 has long been known as a proliferative factor for MM cells. While neoplastic plasma cells can produce IL-6, the most accepted view is that the major source of this cytokine in the BM environment are BM stromal cells, osteoclasts and myeloid precursors cells from the early myeloblast to the intermediate myelocyte maturation stages[57]. The latter population may contain eosinophils, which have been recently reported to support the early growth of murine neoplastic plasma cells in their BM niche[43]. Our data lend further credit to the role of eosinophils as key cells in the early neoplastic plasma cell niche, but do not exclude the role of additional IL-6-producing cells in the BM environment. The role of eosinophils in early MM is further supported by the finding that progression to MM was delayed in early-MM Vk*MYC mice only if treated with the combination of antibodies specific for IL-17, IL-17RA, and IL-5, and therapeutic efficacy correlated with a reduced BM accrual of both Th17 cells and eosinophils. In more advanced MM, as the one reproduced in t-Vk*MYC MM mice, anti-IL5 antibodies are ineffective, and blocking the IL-17 signaling is sufficient to delay MM. Thus, our findings confirm that eosinophils are required for the maintenance of neoplastic plasma cells in the BM niche[43] at the early stage of disease, and add the notion that IL-17 is one critical cytokine in the BM microenvironment that activates eosinophils to release factors supporting neoplastic plasma cells. As the role of IL-5 as growth factor for myeloma plasma cells is debated[43,58], and IL-5 should not impact on BM stromal cells, one mechanism

by which anti-IL17, anti-IL-17RA, and anti-IL5 antibodies acted in Vk*MYC mice is a reduced accrual and survival of eosinophils and consequently of Th17 cells. While our data have highlighted a relevant crosstalk between eosinophils and Th17 cells in the BM of Vk*MYC mice, other cells within the tumor microenvironment produce IL-17, and also stromal cells respond to IL-17 by producing IL-6[32]. Our therapeutic approach should also target these cells.

Currently, the standard of care for patients with SMM has been observation until symptomatic disease occurs because of the limits in predicting disease progression[13]. At least two outputs of this study address this relevant unmet clinical need. Firstly, our data suggest that in patients with SMM a high level of IL-17 in the BM predicts a faster progression to MM. Thus, IL-17 might represent an early biomarker of high-risk SMM patients[13]. One limit of our study is in the number of patients analyzed. A much larger sample size is needed to assess whether IL-17, independently from genetic properties of neoplastic plasma cells, could be a prognostic factor at the specific stage of SMM. Results reported herein should prompt initiating a large multicenter study to address this medical need.

Additionally, the Food and Drug administration has recently approved the use of anti-IL-17A and anti-IL-5 antibodies for the treatment of immune-mediated diseases[59–61]. The availability of these clinical-grade antibodies and our data suggest investigating if targeting the IL-17-eosinophil immune axis would represent a potential treatment for SMM patients at high risk to progress to symptomatic MM.

## Methods

**Patients and BM plasma samples**. Bone marrow (BM) plasma aspirates were obtained from patients fulfilling the International Myeloma Working Group (IMWG) diagnostic criteria after informed written consent, in compliance with all the relevant ethical regulations, and with full ethical approval from the Mayo Clinic institutional review board (authorization #12-001145). Patient's disease staging, collection sample date and MM diagnosis date are reported in Supplementary Table 1. BM plasma samples were obtained by centrifugation of BM aspirates and cryopreserved in the gas-phase of liquid nitrogen

**IL-17 quantification in human BM plasma**. BM plasma samples from the Mayo Clinic Rochester biobank were analyzed with Cytokine Human Magnetic 30-Plex panel for Luminex platform (LHC6003M, Life Technologies, Waltham, MA) and acquired on a Luminex 200 system equipped with with xPONENT 3.1 software (Thermo Fisher Scientific, Waltham, MA).

**Mice**. All mice used in this study were on a C57BL/6 genetic background. WT C57BL/6J mice were purchased from Charles River Breeding Laboratories, Calco IT, or The Jackson Laboratories, Bar Harbor, ME. In Vk*MYC transgenic mice[14] the activation of the transcription factor MYC, whose locus is found rearranged in half human MM tumors[62] including SMM[63], occurs sporadically through the exploitation of the physiological somatic hypermutation process in germinal center B cells. Within a year, although with variable intensity, all mice develop a monoclonal plasmacytosis confined to the BM, a measurable serum M-spike, and progressively show typical endorgan damage[14]. The model has been already validated as a faithful model to predict single agent drug activity in human MM[14,17]. IL-17[KO] mice[29] were kindly provided by Yoichiro Iwakura (Institute of Medical Science, Tokio, Japan). To avoid genetic drifting, Vk*MYC mice were backcrossed into IL-17[KO] mice for at least 6 generations before generating homozygous Vk*MYC IL-17[KO] breeding pairs. Vk*MYC mice were screened by Real Time PCR in order to identify experimental Vk*MYC[+/−] animals with the following primers: primer 1 (5′-ACAGCTACGGAACTCTTGTGCGT-3′), primer 2 (5′-TCAGC-CAAGGTTGTGAGGTTGCA-3′). C57BL/6-Tg(TcraTcrb) 1100Mjb/J (OTII) mice[34] were originally provided by William R. Heath (University of Melbourne, Parkville, Victoria, Australia). Kaede-transgenic mice on a C57BL/6 background were generated by Miwa Yoshihiro at the University of Tsukuba, Japan[26]. All these mice were maintained under specific pathogen-free conditions (i.e., the rodents were housed in isolated rooms, fed sterilized food and water, and routinely tested and determined free of designated pathogens capable of interfering with research objectives; SPF) in the San Raffaele facility and experiments were performed according to state guidelines and approved by the European Community Guidelines (Authorizations #574, #1147, #863). KAEDE-transgenic mice were crossed with IL-17A FP635 reporter knock in mice[27], all on the C57BL/6 background. For photoconversion, the small intestine of anesthetized Kaede/FP635-transgenic mice

was subjected to lighting using a Blue Wave LED Prime UVA (Dymax), essentially as described before[28]. Control mice were sham operated. These animals were maintained under SPF conditions in the Universitätsklinikum Hamburg-Eppendorf facility and treated in accordance with the European Community Guidelines and with the approval of the Universitätsklinikum Hamburg-Eppendorf Institutional Animal Care and Use Committee (authorization # 62/14). The animals reported in Fig. 1a, b, c were bred and maintained in a conventional animal facility (i.e., the rodents were housed in dedicated rooms, and routinely tested for designated pathogens; US1 and US2) at he Mayo Clinic Arizona, under The Mayo Foundation Institutional and Albert Einstein College of Medicine Animal Care and Use Committee approval #A01948. US1 colony belongs to animals generated before 2014 from breeding between littermate Vk*MYC mice. US2 colony belongs to animals generated after 2015 from breeding between Vk*MYC homozygous males and C57BL/6J females purchased from the Jackson lab. Animals within the IT colony were rederived into C57BL/6J mice from the Charles Rivers, and generated in the period 2012–2018 from breeding between Vk*MYC homozygous mice and WT littermates. When appropriate, animal diet was specified in the Results section. Animal facilities were constantly monitored for the presence of relevant pathogens, and resulted free of those pathogens.

**Serum protein electrophoresis.** Mouse blood was periodically collected in Eppendorf by retro-orbital sampling. Semi-automated electrophoresis was performed on the Hydrasys instrument (Sebia, Lissex, France). According to the manufacturer's instructions, 10 μL of undiluted serum were manually applied to the Hydragel agarose gels (Sebia). The subsequent steps: electrophoresis (pH 9.2, 20 W constant current at 20 °C), drying, amidoblack staining, de-staining and final drying were carried out automatically. The use of Hydrasys densitometer and Phoresis software (Sebia) for scanning resulting profiles provided accurate relative concentrations (percentage) of individual protein zones. M-spike levels were calculated as total gamma globulins/albumin ratio (G/A)[17].

**Microbiome analysis.** Bacterial DNA from 50 mg of fecal material was extracted using PowerFecal DNA Isolation Kit (MoBio) following manufacturer's instruction with only one minor modification in lyses time (15 min instead of 5 min) to try to retrieve all difficult-to-lyse bacteria. Purified DNA was quantified and 200 ng per reaction were used to amplify 16S V3-5 regions using barcoded sample-specific primers and FastStart High Fidelity System (Roche) with this thermocycler program: 95 °C for 5 min, 40 cycles of (95 °C for 30", 55 °C for 45" and 72 °C for 1 min) and stored at 4 °C until usage. Amplicons were loaded on 1% agarose gel and purified with QiaQuick Gel Extraction kit (Qiagen) and AMPure XP beads (Beckman Coulter) to remove primer dimers, and used for emulsion-PCR following 454 GS Junior manufacturer's instruction (Roche). Then, emulsion-PCR was purified and captured beads with our correct amplicons were used to load the instruments for the sequencing run. After quality filtering, resulting sequences (>250 bp) were analyzed with QIIME software (1.6.0). Principal component analysis (PCA) was performed on the resulting matrix of unweighted UniFrac distances between samples and statistical analysis was performed on the proportional representation of taxa (summarized to Phyla, Class, Order, Family and Genus levels), using unpaired Student's t-tests.

**Antibiotic treatment and challenge with tumor cells.** Two weeks before I.V. tumor cell challenge ($1 \times 10^6$ Vk12598 cells derived from a MM Vk*MYC mouse[14]), Ciprofloxacin (300 mg/L) and Metronidazole (1 g/L; Sigma-Aldrich), known to eliminate the majority of intestinal bacteria[64], were added to the drinking water of 8–10 week old WT or IL-17$^{KO}$ C57BL/6J recipients, and mice were maintained on antibiotics throughout the duration of the experiment. Mice were monitored for M-spike appearance as described above, and sacrificed within 70 days. Vk12598 cells were generated in Bergsagel lab, and were not authenticated. As these cells do not grow in vitro, they were not tested for mycoplasma contamination.

**Bacteria cultivation and mice infection.** *P. heparinolytica* DSM 23917 and *P. melaninogenica* DSM 7089 (DSMZ, Germany) were cultured in Brain Heart Infusion (BHI) medium at 37 °C under anaerobic conditions, following manufacturer's instructions. Fifty microliter of bacterial growth where then transferred on chocolate agar plates and cultivated at 37 °C in AnaeroJar 2.5 L Jar System (OXOID) and using AnaeroGen 2.5 L (Thermo Scientific) in order to generate an anaerobic atmosphere. To infect mice with the selected *Prevotella* strains, Ampicillin (1 g/L; Sigma-Aldrich), Vancomycin (0.5 g/L; Sigma-Aldrich), and Neomycin (1 g/L; Sigma-Aldrich) were added to the drinking water of 6 week old WT C57BL/6J and Metronidazole (2 mg/mouse; Sigma-Aldrich) was administered by oral gavage 3 times per week. Two weeks later, antibiotic-treated animals were infected by gavage with with *P. heparinolytica* or *P. melaninogenica* for 3 consecutive days/week, until the end of the experiment. Each recipient mouse received an oral gavage of 200 μL. *Prevotella* in the stool of infected mice was confirmed by RT-PCR. After two weeks of bacteria infection, mice were challenged I.V. with $1 \times 10^6$ Vk12598 cells. For disease monitoring, mouse blood was collected by retro-orbital sampling once a week starting from the third week since tumor challenge, and analyzed by Serum Protein Electrophoresis as described above.

**Antibody treatments.** αIL-5 (Clone TRFK5, BioxCell), or αIL-17A (Clone P59234.19, Amgen) and αIL-17R (Clone PL-31280, Amgen) or αIL-17A, αIL-17R and αIL-5, or isotype control (GL117, rat IgG2a) were injected i.p. (once a week for 9 weeks, 150 μg of each monoclonal antibody per mouse) in Early-MM Vk*MYC mice. Every three weeks mice were bled for M-spike quantification. Five days after the last injection mice were sacrificed and their BM assessed for the presence of Th17 cells and eosinophils. In experiments with the Vk12598 murine cell line, sex- and age-matched C57BL6J or IL-17$^{KO}$ mice were challenged i.v. with $1 \times 10^6$ Vk12598 cells, and C57BL6J mice were weekly injected i.p. with 100 μg per mouse of αIL-5 (Clone TRFK5, BioxCell), or αIL-17A (Clone P59234.19, Amgen) and αIL-17R (Clone PL-31280, Amgen), their combination, or isotype control (GL117, rat IgG2a) starting from the week of tumor challenge. Every week mice were bled for M-spike quantification.

**Collection of BM serum and cells.** Each femur devoid of epiphyses was placed into a 0.5 mL eppendorf tube whose bottom was pierced with a 16 G needle. The pierced eppendorf tube containing the bone was subsequently placed into a 1.5 mL eppendorf tube and centrifuged (Heraeus$^{TM}$ Pico$^{TM}$ 17 Microcentrifuge, ThermoFisher Scientific, Waltham, MA USA) for few seconds. The BM pelleted material, containing both serum (approximately 10 μL) and cells was resuspended in 100 μL PBS, and used for flow cytometry analyses. Alternatively, the resuspended material was centrifuged again to separate diluted serum from cells, and stored at −80 °C.

**Flow cytometry.** Peyer's Patches were removed from the small intestine, and gently disaggregated with the help of tweezers. BM cells from the same animals were collected as described above. Single cell suspensions were labeled with fluorochrome-conjugated monoclonal antibodies (either from BD Bioscience, Buccinasco IT, Biolegend Europe, Uithoorn The Netherlands, or eBioscience Inc, Prodotti Gianni, Milan, IT, or R&D Systems, Space Import-Export srl, Milano, Italy) after neutralization of unspecific binding with FcR blocker (BD Biosciences), and acquired by BD LSR Fortessa™(BD Biosciences). The antibodies used were: αIL17A (clone TC11-18H10, cat 559502), αIL17RA (clone PAJ-17R cat 17-7182-80), α α4B7 (clone DATK32, cat 120607), αCD3 (clone 145-2C11, cat 100330), αCD8 (clone 53-6.7, cat 560776), αCD4 (clone GK1.5, cat 100536), αNK1.1 (clone PK136, cat 108705), αCD90.2 (clone 30-H12, cat 105324), αCD138 (clone 281-2, cat 553714), αpSTAT3 (clone pY705, cat 557815), αIL6 (clone MP5-20F3, cat 561367), αSiglecF (clone E50-2440, cat 562068), αCD45 (clone 30-F11, cat 561487), αLy-6G (clone 1A8, 561236), αLy6C (clone HK1.4, cat 128017), αCD11b (clone M1/70, cat 101224), αCD127 (clone A019D5, cat 351303), Lin (clone 17A2/RB6-8C5/RA3-6B2/Ter-119/M1/70, cat 133301), αCD11c (clone N418, cat 117318), α$^-$Ab (clone 25-9-17, cat 114406), either from BD Bioscience, Biolegend Europe, Uithoorn The Netherlands, or eBioscience Inc, Prodotti Gianni, Milan, IT, polyclonal αIL17RC (cat FAB2270A) from R&D Systems, Space Import-Export srl, Milano, Italy). For surface staining all antibodies were diluted 1:200, with the exception of αIL17RC diluted 1:20, for intracellular staining antibodies were diluted 1:100. Data were analyzed using the FlowJo software (TreeStar Inc, Ashland, OR, USA). Cells were also assessed for intracellular cytokine production after 6 h at 37 °C of stimulation with Phorbol Myristate Acetate (PMA)/ionomycin. GolgiPlug® (BD Bioscience) was added to the samples during the last 5 hours of culture. After incubation, cells were washed and stained for surface markers 15 min at 4 °C, fixed and permeabilized with Fixation/Permeabilization Kit (BD-bioscience). Cells were then washed and stained for intracellular markers 30 min at 4 °C and acquired by FACS (BD LSR Fortessa™). Data were analyzed using the FlowJo software (Treestar Inc).

**Th17 polarization in vitro.** OTII splenocytes were cultured in complete IMDM for 7 days under stimulation with anti-CD3/CD28 Dynabeads ($4 \times 10^5$ beads/$2 \times 10^5$ cells; Invitrogen, Thermo Fisher, Milan, IT), and in the presence of the combination of IL-6 (20 ng/mL, PeproTech, tebu-bio, Milan, Italy), TGF-β1 (2 ng/mL, R&D Systems, Minneapolis, MN), anti-IL-4 (10 μg/mL, cat 554432, BD Biosciences) and anti-IFN-γ antibodies (10 μg/mL, cat 517904, Biolegend). Alternatively, stimulated cells were cultured in the presence of BM serum (1:25 final dilution). After 7 days OTII cells were tested for intracellular cytokine production by flow cytometry.

**Intracellular phospho-protein analysis by flow cytometry.** Vk*MYC plasma cells were stimulated with recombinant mouse IL-6 (100 ng/mL, 30 min), or IL-17 (50 ng/mL, 20 h; Proleukin), or with BM serum (1:20) with or without the addition of anti-IL-17A antibodies (clone: TC11-18H10.1, 3 μg/well, Biolegend) respectively, and subsequently fixed by Cyto-Fix buffer (BD Biosciences), and permeabilized in Perm Buffer III (BD Biosciences) on ice. Staining was performed by anti-STAT3 (pY705, cat 557815, BD Biosciences) Alexa Fuor 647-conjugated antibodies and analyzed by flow cytometry.

**In vitro induction of mouse bone marrow-derived eosinophils.** Eosinophils were obtained from BM precursors as described in the ref.[41]. In brief, BM cells were collected from the femurs and tibiae of WT C57BL/6J mice by flushing the opened bones with PBS (Euroclone, Pero, Italy). The BM cells were cultured at $10^6$/mL in medium containing RPMI 1640 (Invitrogen) with 20% FBS (Cambrex), 100 IU/mL

penicillin and 10 μg/mL streptomycin (Cellgro), 2 mM glutamine (Invitrogen), 25 mM HEPES and 1x nonessential amino acids and 1 mM sodium pyruvate (Life Technologies), and 50 μM 2-ME (Sigma-Aldrich) supplemented with 100 ng/mL stem cell factor (SCF; PeproTech) and 100 ng/mL FLT3 ligand (FLT3-L; Pepro-Tech) from days 0 to 4. On day 4, the medium containing SCF and FLT3-L was replaced with medium containing 10 ng/mL recombinant mouse IL-5 (R&D Systems) only. On day 8, the cells were moved to new flasks and maintained in fresh medium supplemented with rmIL-5. BM eosinophils were stimulated with BM serum (1:20) with or without the addition of anti-CCR3 (CD193, Clone: J073E5, 3 μg/well, cat 144503, Biolegend) or anti-IL-17A antibodies (clone: TC11-18H10.1, 3 μg/well, cat 506902, Biolegend).

**BM serum cytokine quantification in mice**. Cytokines were quantified by the Myriad RBD™ multiplex immunoassay (Myriad RBD, Austin, TX, USA). The sera were 1:10 diluted with PBS, and stored at −80 ˚C until sending to Myriad RBD for cytokine quantification.

**Statistics analyses and reproducibility**. Sample size was chosen taking into account the means of the target values between the experimental group and the control group, the standard error and the statistical analysis used. Based on our previous experience[14,17,33] and preliminary data obtained in the Vk*MYC and t-Vk*MYC MM models, we estimated a number of 5 and 10 animals per experimental group for the in vitro and in vivo experiments, respectively, to ensure adequate power (alfa = 0.05 and power = 0.80) to detect significant variations in the measured events. No samples or animals were excluded from the analyses. Grubb's test was applied to exclude outliers. Animals were always matched for sex and age. Randomization was performed for in vivo experiments assessing the therapeutic efficacy of antibodies. No blinding was done for in vivo experiments. Data were analyzed with GraphPad Prism version 7. The data are presented as mean ± standard deviation of the mean, individual values as scatter plot with column bar graphs and were analyzed using Student's $t$-tests (paired or unpaired according to the experimental setting) by a two-sided and, when indicated, followed by Wilcoxon post-test. One-way ANOVA was used to compare three or more groups in time point analyses. Differences were considered significant when $P < 0.05$ and are indicated as NS, not significant, $*P < 0.05$, $**P < 0.01$, $***P < 0.001$. Non-parametric tests were applied when variables were not normally distributed using the SPSS statistical software. $N$ values represent biological replicates. Survival curves were compared using the log-rank test (Mantel–Cox). All the statistics and reproducibility are reported in the figure legend. Relevant data are available from the authors.

## Data availability

The authors declare that the data supporting the findings of this manuscript are available within the paper and its supplementary information. Gene expression profiling data of primary SMM cells were obtained from ref. [65]. The probe set used for IL-17RA expression was 207707_at. Datasets were analyzed by Student's $t$-test directly at www.oncomine.org as 207707_at. All other remaining data supporting the findings of this study are available from the authors upon reasonable request.

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

## Acknowledgements

We are indebted to Yoichiro Iwakura (Institute of Medical Science, Tokio, Japan) and William R. Heath (University of Melbourne, Parkville, Victoria, Australia) for providing us IL-17KO and OT-II breeders, respectively. We thank Maria Protti (San Raffaele Scientific Institute, Milan, Italy) for critically revising the manuscript. Amgen Inc. (Thousand Oaks, CA, USA) kindly provided us with anti-IL17RA and anti-IL-17 antibodies. We thank Caligaris-Cappio (San Raffaele Scientific Institute, Milan, Italy) for his intellectual contribuition to the project. The work was supported by Associazione Italiana per la Ricerca sul Cancro (AIRC), AIRC 5 × 1000 Molecular Clinical Oncology Special Program, Milan, IT (grant no. 9965 to M.B. and G.T.). This work was also supported by research grants from the National Cancer Institute: CA190045 (M.C.), CA186781 (M.C.), CA168762 (V.S.R.), CA186781 (P.L.B.), and CA195688 (P.L.B.). A.C. was awarded a fellowship from AIRC/FIRC and A.C. and A.B. conducted this study in partial fulfillment of their Ph.D. at San Raffaele University.

## Author contributions

M.B. and A.C. developed the concept of the study. M.B., A.C., A.B., and M.C. designed and conceived the experiments. A.C., A.B., M.C., R.F., M.G., L.G.P., V.M.G., and M.E.S. performed the experiments. M.G. and K.J.H. helped with large in vivo experiments. S.H. and L.G.P. designed and analyzed the experiments with Kaede-transgenic mice. M.G. and M.E.S. took care of the mouse colonies. G.T. analyzed cytokine data. S.K., K.J.H., and V.S. R. provided patients' sera. M.T., Y.M., E.E., and R.A.F. provided genetically modified mice. A.C., A.B., and M.C. prepared the figures and tables. M.B. and A.C. wrote and prepared the manuscript. A.B., M.C., R.F., L.G.P., S.H., F.C., and P.L.B. commented on the manuscript. All the authors read and approved the manuscript.

## Additional information

**Competing interests:** The authors declare no competing interests.

