## [Peer Review File · Nature Communications]

Reviewers' comments:

Reviewer #1: Expert in MM and SMM
(Remarks to the Author):

The paper by Calcinotto et al, "Microbiota-driven interleukin-17-producing....." furnishes evidence that microbiota promotes differentiation of Th17 cells in the gut and that these cells migrate to the bone marrow where they support the growth of multiple myeloma (MM) cells via a mechanism that involves bone marrow eosinophils. The study involves for the most part the Vk*MYC model but a few experiments are also performed with patients with smoldering MM. The experimental work is generally well done and supports the conclusions of the paper. The paper is generally well-written.

Comments

- 1) The influence of Th17 and eosinophils on development of MM has been described previously, as referenced by the authors (in this context, ref. 42 describes the influence of eosinophils in a mouse model for MM, and credit should be given to this reference for demonstrating this, rather than only citing it for a technique to experimentally remove eosinophils). This having been said, the authors extend these previous findings to another mouse model, and the experiments are comprehensive.
- 2) There are alternative opinions on the niche that supports MM development in the bone marrow, and these alternative views might be briefly mentioned (e.g. the work of Huard et al).
- 3) The influence of gut microbiota, via Th17 and eosinophils, is deemed more original and the data appear convincing. I could not see any mentioning of the old work by Potter et al. on Mineral Oil-Induced Myelomas (MOPC). Could it be that injection of mineral oil in the peritoneal cavity could induce Th17 cells, thus favoring MM development? The authors may briefly touch upon this issue.
- 4) The last parts of Fig. 2 and the Fig. 2 legend do not correspond with each other.
- 5) The heat maps are not very convincing the way they are presented. Moreover, a scale should be included.
- 6) in Fig. 4, BC, MM-Before treatment and MM-After treatment are not introduced in the Fig. legend. Please do so, it makes it easier for the reader.

Reviewer #2 : Expert in Microbiota and Cancer
(Remarks to the Author):

The manuscript by Calcinotto et al., reports an interesting connection between multiple myeloma progression and microbiota-induced IL-17, as well as potential synergistic role for BM eosinophils (Eo). Although a role for IL-17 in disease pathogenesis has been proposed the manuscript makes several novel observations. In particular, it describes an increase in IL-17 effects in early disease and proposes that IL-17 may be an early marker for increased disease progression. Another novelty is that microbiota composition is proposed to drive the pathogenic IL-17. Furthermore the authors propose that IL-17 promotes pathogenic Eo activation, which exacerbates disease progression. In general, the study is of significant interest and importance. It makes several interesting observations and proposes an attractive network of interactions that will lead to further interesting studies. There are, however, several major questions and experiments that have to be addressed/performed to support the conclusions.

Major points

The role of microbiota is an important observation of the study. However, although role for microbiota is suggested by the experiments, it is not directly demonstrated and therefore not proven. The fact that mice from two different facilities have vastly different responses is very interesting, but is not sufficient to prove a role for microbiota. Details for the two animal facilities and how mice were matched for experiments from these two facilities are not provided. Mice from different facilities are expected to have different microbiota, but they are also expected to differ in

many other environmental conditions, e.g. housing conditions, diet, presence of bacterial and non-bacterial intestinal pathogens. Role for microbiota can be directly examined by co-housing experiments, fecal transplantation, defined bacterial colonizations, gnotobiotic animals, etc. At least some of these experiments need to be performed in order to claim role for microbiota. Antibiotics experiments are not sufficient to prove role for microbiota, because Abx may have indirect effects. Moreover, Abx experiments were performed in a different disease model.

Although IL-17 is shown to activate Eo in vitro, there is no data to show that this occurs in vivo. This can be examined for example by looking at Eo activation in IL-17-KO mice. Similarly the role of Eo in MM in the model is unclear, because aIL-5 only treatment data are not presented.

The term IL-17-Eo axis needs to be reconsidered or explained more carefully. It suggests that IL-17 activates Eo and this is required for disease progression. However, the blocking experiments seem to support non-redundant roles for IL-17 and Eo (no difference after only aIL-17). Is there an effect on Eo activation following IL-17 blockade? What happens after aIL-5 treatment only? If there is an IL-17-Eo axis, then Eo depletion will affect disease and will also decrease the differences between Wt and IL-17-KO mice.

Methodologies and statistics. Experimental details are missing for a number of experiments and some experiments are performed only once and contain insufficient amount of animals to make robust conclusions. Some examples are listed below.

Specific Points

Microbiota:

How were mice from the two facilities matched in experiments (age, sex?).

It is crucial that microbiota transfer experiments are performed, so littermate mice that differ only in their microbiota are compared.

Were the experiments on IL-17KO mice performed with littermate controls? Where were these animals (and the controls) housed? Because of the possible effects of microbiota it is crucial that genetic experiments are performed with littermate controls.

IL-17+ cell increase in conventional facility and sick mice (e.g. Fig 1G, 1E). This needs to be documented better. Frequency of total live cells is reported, which makes it unclear of whether this is due to increase in IL-17+ cells or decrease in another cell subset in the PP/BM. Total numbers or normalization to a reference population should be provided. What is the nature of these cells? Are they Th17 cells? If yes, frequencies within CD4 T cells can be presented.

The Kaede experiments mainly follow Th17 cells. Is this the main proposed IL-17 producing subset in the BM? Is colonization with Th17 cell-inducing bacteria, e.g. SFB or Prevotella, sufficient to promote disease?

Fig 1N – it seems that there is a difference between IL17KO and IL-17KO+Abx at the first timepoint. This experiment is important to show dependence on IL-17, but only 5 mice/group in a single experiment are presented and statistics are missing. WT+ABX condition is also missing. More experiments/animals are needed. Similarly on Fig 1O the animal numbers are low in a single experiment. Moreover, it seems that the penetrance of disease is low in the controls (3 out of 5 mice do not show disease). Why is that? Also, can the authors clarify what is plotted on this figure? What is % of M-spike?

Abx treatment – when do M-spikes appear in treated mice? How delayed is the appearance of M-spike and what is the difference in quantification of M-spike? Kinetics similar to the ones in Fig 1A/2A and 2B can be presented.

Fig 1E – the vehicle graph is incomplete. Do any of the animals survived beyond 45 days?

Legend for Figure 2D-G is wrong

Suppl Table 1 is missing

Figure 4C is impossible to understand. It should be removed or replaced with a more relevant analysis.

For some figures it will be helpful to include representative FACS plots in order to understand better the proposed phenotypes/changes. For example on Figure 5 where IL-6 production is measured, especially because MFI changes are used. MFI changes could be supported by a more reliable method for protein quantification, e.g. ELISA

Fig 5F – this figure is difficult to read. Maybe some statistics and quantification data for MCP-3 expression under different treatment groups, rather than a heat map, would be better.

Missing or incomplete Methods:

Description of the two animal housing facilities is missing

The nature of the controls used should be noted for all experiments. Are they littermates, age, sex-matched, etc.?

What is meant by “Vk*MYC mice were backcrossed into IL-17KO mice for 6 generations”? Are Vk*MYC, IL-17KO and Kaede mice on B6 background?

M-spike quantification method is not described.

How was early and late stage defined on Fig 2?

How were PP single cell suspensions prepared?

In vitro polarization – more detail is needed. What is BM serum and how was it prepared?

We thank the Editor and the reviewers for their positive comments and constructive criticisms that helped to further improve the quality of our manuscript. We have now reorganized our manuscript on the basis of a number of critical new findings. Furthermore, we have addressed in full the critiques and constructive suggestions of the reviewers, which have tremendously helped us at strengthening and corroborating the original conclusions, specifically, the relevance of the microbiota and the IL17-Eosinophils axis.

**Reviewer #1: Expert in MM and SMM
(Remarks to the Author):**

The paper by Calcinotto et al, "Microbiota-driven interleukin-17-producing....." furnishes evidence that microbiota promotes differentiation of Th17 cells in the gut and that these cells migrate to the bone marrow where they support the growth of multiple myeloma (MM) cells via a mechanism that involves bone marrow eosinophils. The study involves for the most part the Vk*MYC model but a few experiments are also performed with patients with smoldering MM. The experimental work is generally well done and supports the conclusions of the paper. The paper is generally well-written.

We thank the Reviewer #1 for her/his constructive comments, which helped to further improve the quality of our manuscript. In this rebuttal, and in our revised manuscript, we have incorporated the her/his suggestions to strengthen the relevance of our previous findings. We have also discussed in the manuscript the literature suggested by Reviewer #1.

Comments

1) The influence of Th17 and eosinophils on development of MM has been described previously, as referenced by the authors (in this context, ref. 42 describes the influence of eosinophils in a mouse model for MM, and credit should be given to this reference for demonstrating this, rather than only citing it for a technique to experimentally remove eosinophils). This having been said, the authors extend these previous findings to another mouse model, and the experiments are comprehensive.

We thank Reviewer #1 for her/his positive comments. The work by Wong and colleagues has been fully acknowledged in the revised version of the manuscript. More in details, the Wong's paper (ref. 41) has been cited at page 14 (*"Indeed, treatment with anti-IL-5 antibodies has been shown to reduce eosinophil numbers in blood and BM of mice⁴¹."*), at page 18 (*"Commensal microbes are not unique in favoring the expansion of pathogenic Th17 cells in MM. As an example, mineral oil, which is used in food, cosmetics and biomedicine, has been reported to promote plasma cells neoplasms in BALB/c mice⁵¹, through IL-6⁵², eosinophils⁴¹, and possibly the expansion of Th17 cells⁵³. Thus, we speculate that several environmental factors in addition to the gut microbiota substantially influence MM progression by inducing pathogenic Th17 cells"*), and at page 19 (*"While neoplastic plasma cells can produce IL-6, the most accepted view is that the major source of this cytokine in the BM environment are BM stromal cells, osteoclasts and myeloid precursors cells from the early myeloblast to the intermediate myelocyte maturation stages⁵⁵. The latter population may contain eosinophils, which have been recently reported to support the early growth of murine neoplastic plasma cells in their BM niche⁴¹."*).

2) There are alternative opinions on the niche that supports MM development in the bone marrow, and these alternative views might be briefly mentioned (e.g. the work of Huard et al).

We thank Reviewer #1 for this suggestion that has helped us to further improve the discussion of our work. We have included the following sentences to the discussion (page 18): *"IL-6 has long been known as a proliferative factor for MM cells. While neoplastic plasma cells can produce IL-6,*

*the most accepted view is that the major source of this cytokine in the BM environment are BM stromal cells, osteoclasts and myeloid precursors cells from the early myeloblast to the intermediate myelocyte maturation stages⁵⁵. The latter population may contain eosinophils, which have been recently reported to support the early growth of murine neoplastic plasma cells in their BM niche⁴¹. Our data lend further credit to the role of eosinophils as key cells in the early neoplastic plasma cell niche, but do not exclude the role of additional IL-6-producing cells in the BM environment. The role of eosinophils in early MM is further supported by the finding that progression to MM was delayed in early-MM *Vk*MYC* mice only if treated with the combination of antibodies specific for IL-17, IL-17RA and IL-5, and therapeutic efficacy correlated with a reduced BM accrual of both Th17 cells and eosinophils. In more advanced MM, as the one reproduced in *t-Vk*MYC* MM mice, anti-IL5 antibodies are irrelevant, and blocking the IL-17 signaling is sufficient to delay MM”.*

3) The influence of gut microbiota, via Th17 and eosinophils, is deemed more original and the data appear convincing. I could not see any mentioning of the old work by Potter et al. on Mineral Oil-Induced Myelomas (MOPC). Could it be that injection of mineral oil in the peritoneal cavity could induce Th17 cells, thus favoring MM development? The authors may briefly touch upon this issue.

To comply with this excellent suggestion, we added the following paragraph to the manuscript (page 17): *“Commensal microbes are not unique in favoring the expansion of pathogenic Th17 cells in MM. As an example, mineral oil, which is used in food, cosmetics and biomedicine, has been reported to promote plasma cells neoplasms in BALB/c mice⁵¹, through IL-6⁵², eosinophils⁴¹, and possibly the expansion of Th17 cells⁵³. Thus, we speculate that several environmental factors in addition to the gut microbiota substantially influence MM progression by inducing pathogenic Th17 cells”.*

4) The last parts of Fig. 2 and the Fig. 2 legend do not correspond with each other.

We apologize for this mistake. In the new version of our manuscript we have corrected the mistake.

5) The heat maps are not very convincing the way they are presented. Moreover, a scale should be included.

We agree with both reviewers that the heat maps were not clear enough to be shown in the figures of the manuscript. This having been said, we are still convinced they contained relevant information for the comprehension of the work. Thus, we modified and included them in the supplementary material (Suppl. Figs. 5 and 6).

6) in Fig. 4, BC, MM-Before treatment and MM-After treatment are not introduced in the Fig. legend. Please do so, it makes it easier for the reader.

We modified the figure legend as follows (page 34): *“Fig. 4 IL-17 levels are increased in the BM of SMM patients rapidly progressing to MM. a mRNA expression of IL-17RA in primary SMM cells of a cohort of 12 newly-diagnosed patients and 22 matched controls (bone marrow) described in ref.⁶³. The expression pattern for the probe set 205707_at is shown. Statistical analysis (Student t test) is reported. b IL-17 levels in the BM plasma of SMM patients that progressed to MM within 3 years since the diagnosis (i.e., < 3 years), or did not progress to MM in the same time frame (i.e., > 3 years). Each dot represents an individual patient. (SMM-Progression >3 n=12, SMM-Progression <3 n=22, MM-Before treatment n=12, MM-After treatment n=11). Data are reported as mean ± SE. Unpaired t test: *P <0.05”.*

**Reviewer #2 : Expert in Microbiota and Cancer
(Remarks to the Author):**

The manuscript by Calcinotto et al., reports an interesting connection between multiple myeloma progression and microbiota-induced IL-17, as well as potential synergistic role for BM eosinophils (Eo). Although a role for IL-17 in disease pathogenesis has been proposed the manuscript makes several novel observations. In particular, it describes an increase in IL-17 effects in early disease and proposes that IL-17 may be an early marker for increased disease progression. Another novelty is that microbiota composition is proposed to drive the pathogenic IL-17. Furthermore the authors propose that IL-17 promotes pathogenic Eo activation, which exacerbates disease progression. In general, the study is of significant interest and importance. It makes several interesting observations and proposes an attractive network of interactions that will lead to further interesting studies. There are, however, several major questions and experiments that have to be addressed/performed to support the conclusions.

We thank Reviewer #2 for having underlined the novelties contained in the manuscript. We also thank the referee for her/his constructive suggestions, which have helped to further improve the quality of our manuscript. In this rebuttal, and in our revised manuscript, we have incorporated all the suggestions of Reviewer #2 to reinforce the relevance of our previous findings. As suggest by Reviewer #2, we have directly demonstrated the role of the microbiota in MM progression by performing defined bacterial colonization experiments. In addition to this, and in order to better dissect the contribution of Th17 cells and eosinophils, we have performed the additional analyses and experiments suggested by Reviewer #2 that we have included in the revised paper.

Major points

The role of microbiota is an important observation of the study. However, although role for microbiota is suggested by the experiments, it is not directly demonstrated and therefore not proven. The fact that mice from two different facilities have vastly different responses is very interesting, but is not sufficient to prove a role for microbiota. Details for the two animal facilities and how mice were matched for experiments from these two facilities are not provided. Mice from different facilities are expected to have different microbiota, but they are also expected to differ in many other environmental conditions, e.g. housing conditions, diet, presence of bacterial and non-bacterial intestinal pathogens. Role for microbiota can be directly examined by co-housing experiments, fecal transplantation, defined bacterial colonizations, gnotobiotic animals, etc. At least some of these experiments need to be performed in order to claim role for microbiota. Antibiotics experiments are not sufficient to prove role for microbiota, because Abx may have indirect effects. Moreover, Abx experiments were performed in a different disease model.

We apologize for the lack of clarity regarding the two animal facilities and how mice were matched for experiments. We went through the entire manuscript to better clarify this issue. All the animals used for the study were sex and aged-matched littermates. When needed, this has been specified in the text. We also modified the nomenclature of the mice in the different experiments to better clarify whether they were affected by *de novo* (Vk*MYC) or transplanted (t-Vk*MYC) MM.

Mice were fed a very similar diet in the two animal facilities. This was specified in the original manuscript at page 7: “*Of note, changes in the microbiota were not apparently related to the diet, because mice in the CNV facility were fed a diet (i.e., PicoLab[®] Rodent Diet 20 5053; LabDiet), whose content of nutrients, minerals, vitamins and calories was comparable to the diet of SPF mice (Teklad Global 18% Protein Rodent Diet; Harlan)*”.

Animals facilities were constantly monitored for the presence of relevant pathogens, and resulted pathogen-free.

To follow the referee's suggestions, in the revised version of the manuscript we have modified the text as follows: (page 7) "*Of note, health reports confirmed the absence of relevant pathogens in the two animal facilities, and changes in the microbiota were not apparently related to the diet, because mice in the CNV facility were fed a diet (i.e., PicoLab[®] Rodent Diet 20 5053; LabDiet), whose content of nutrients, minerals, vitamins and calories was comparable to the diet of SPF mice (Teklad Global 18% Protein Rodent Diet; Harlan)*". We also added similar details in the Materials and Methods section.

We thank the Reviewer for having suggested additional experiments to prove a role for the gut microbiota in MM. The new experiments are now reported in Fig. 1e and in the related text: (page 7) "*To further support the link between gut microbiota and MM progression, antibiotic-treated SPF AF mice housed in an isolator were subjected to fecal microbiota transplantation (FMT) with stools from SPF AF mice, or administration of *Prevotella heparinolytica*, the *Prevotellaceae* mostly represented in the CNV AF (Fig. 1c and data not shown), before challenge with *Vk12598* cells. As expected, FMT of SPF AF mice with stools from SPF AF animals did not impact MM progression and animal survival, whereas disease-related death was accelerated in *P. heparinolytica*-infected *t-Vk*MYC* MM mice (Fig. 1e). Thus, in these experimental conditions, microbiota constituents, and *P. heparinolytica* in particular, is necessary for the generation of a microenvironment prone to tumor cell engraftment and expansion*". We also commented these results in the Discussion section: (page 17) "*Prevotellaceae, which are known to promote Th17 differentiation locally and at distant sites²², were almost only present in CNV animals, and *P. heparinolytica* accelerated MM progression. Thus, *Prevotella* species are primary suspects also in humans, in which the increased abundance of these bacteria at mucosal sites has been associated with Th17-mediated diseases including periodontitis²² and rheumatoid arthritis⁴⁶. Interestingly, in the humanized HLA-DQ8 murine model, *P. melanogenica* and *P. histicola* augmented and suppressed rheumatoid arthritis, respectively⁴⁷, and *P. histicola* suppressed experimental autoimmune encephalomyelitis by modulating IL-17 production⁴⁸. On this line, an increased abundance of *Prevotella* species has been associated with reduced intestinal Th17 cell frequency and high disease activity in multiple sclerosis⁴⁹. All together, these findings suggest that selected members of the same genus have different disease modulating properties*". These relevant new finding has also been incorporated in the Abstract.

Although IL-17 is shown to activate Eo in vitro, there is no data to show that this occurs in vivo. This can be examined for example by looking at Eo activation in IL-17-KO mice. Similarly the role of Eo in MM in the model is unclear, because aIL-5 only treatment data are not presented.

To address this interesting point, we analyzed the frequency of eosinophils and their functional capability in *Vk*MYC IL17^{KO}* mice developing *de novo* MM, and compared them to MM *Vk*MYC IL17^{WT}*. In accordance with our hypothesis and results from our *in vitro* experiments, the constitutive absence of IL17 limited the accumulation of eosinophils in the BM and also their ability to produce IL6. New data are incorporated in Fig. 5 and in the text: (page 12) "*Eosinophils were indeed present in the BM of *Vk*MYC IL-17^{WT}* mice developing *de novo* MM, and their frequency increased with disease progression (Fig. 5a). Interestingly, eosinophils were not increased in the BM of MM *Vk*MYC IL-17^{KO}* mice (Fig. 5a). When these cells were assessed for cytokine production, which is a marker of activation, increased frequency of IL-6⁺ eosinophils (Fig. 5b) were found in the BM of Early- but not Late-MM *Vk*MYC IL-17^{WT}* mice (Fig. 5c). Again, the lack of IL-17 prevented accumulation of eosinophils in the BM of *Vk*MYC IL-17^{KO}* mice affected by MM (Fig. 5c). Finally, the eosinophil mean fluorescence intensity (MFI) for IL-6 was also increased in Early-MM *Vk*MYC* mice (Fig. 5d), thus suggesting that eosinophil activating factors were enriched in the BM of these mice, particularly at early phases of disease*".

We also tested the effects of anti-IL-5 antibodies on disease progression and BM accrual of eosinophils in *t-Vk*MYC* MM mice. New data have been incorporated in Suppl. Fig. 8 and in the

text at page 14: “Treatment with only anti-IL5 antibodies, while associated with reduced BM accrual of eosinophils (Suppl. Fig. 8c and d), neither impacted disease progression in our MM models (Fig. 6b and Suppl. Fig. 8a), nor affected Th17 accrual in the BM of t-Vk*MYC MM mice (Suppl. Fig. 8b). All together, these data support the concept that disease progression in Vk*MYC mice is propelled by the IL-17-eosinophil axis, which can be broken by the combination of cytokine-specific antibodies (Fig. 7)”.

The term IL-17-Eo axis needs to be reconsidered or explained more carefully. It suggests that IL-17 activates Eo and this is required for disease progression. However, the blocking experiments seem to support non-redundant roles for IL-17 and Eo (no difference after only aIL-17). Is there an effect on Eo activation following IL-17 blockade? What happens after aIL-5 treatment only? If there is an IL-17-Eo axis, then Eo depletion will affect disease and will also decrease the differences between Wt and IL-17-KO mice.

We thank Reviewer #2 for having raised this point, and we made our best to clarify this issue in the revised version of the manuscript. Experiments with blocking antibodies have been conducted both in the Vk*MYC mice and the transplanted t-Vk*MYC MM model. Data obtained in the primary Vk*MYC model suggest that IL-17 and eosinophils exert a synergistic and non-redundant activity in the early phases of MM.

We have shown that Th17 cells were enriched in the BM of Early-MM mice, and disease appearance and progression to the symptomatic phase were substantially delayed in Vk*MYC IL-17^{KO} mice (Figs. 2a and b). Additionally, the BM of Early-MM mice contained enough IL-17 to induce STAT3 phosphorylation in neoplastic plasma cells (Fig. 3). All together, these data strongly support a pathogenic role for IL-17 in the early phases of disease in Vk*MYC mice.

Nonetheless, treatment with anti-IL17 and anti-IL17R antibodies neither affected the number of BM infiltrating Th17 cells and eosinophils, nor impacted disease progression in Vk*MYC mice (Fig 6), suggesting that additional mechanisms are required at this stage of disease to synergise with the pro-tumor activity of IL17. These results are at odds with a previous report showing that IL17 produced by human neoplastic plasma cells was needed for proper MM development in immunodeficient mice (Prabhala R.H et al. Leukemia 2015). We obtained similar results in the t-Vk*MYC MM model. Indeed, treatment with anti-IL17 and anti-IL17R antibodies delayed MM development in these mice (Fig. 1o). Results obtained in the two transplantable models are in line with *in vitro* data obtained with human neoplastic plasma cells collected from symptomatic MM patients (refs. Noonan K. et al. Blood 2010; Dhodapkar K.M. et al. Blood 2008; Prabhala R.H. et al. Blood 2010). All together, these data suggest that in the early phases of disease other factors cooperate with IL-17 in supporting plasma cell survival and proliferation.

We also found that activated eosinophils were enriched in the BM of Early-MM mice in the Vk*MYC model (Fig. 5). More importantly, we report that mouse eosinophils express a functional IL-17R (Fig. 5e), and the BM of Early-MM mice contain enough IL17 and MCP3 to activate eosinophils (Fig. 5h and i). However, treatment with anti-IL5 antibodies did not alter MM progression in Early-MM mice (Fig. 6), suggesting that also eosinophils required an additional factor to exert their pro-tumor activity in MM. Therefore, we combined anti-IL16 and anti-IL17R with anti-IL-5 antibodies. As hypothesized, the triple combined treatment significantly reduced the accumulation of both Th17 and eosinophils in the BM of Vk*MYC, and delayed disease progression (Fig 6).

Originally, we did not investigate the BM of IL-5-treated Early-MM mice because the treatment did not impact disease progression (Fig. 6b). We agree with Reviewer #2 that it was important to assess the effects on the BM of anti-IL5 antibodies when given as single treatment. Treatment was conducted in transplantable t-Vk*MYC MM model, and the results are reported in Suppl. Fig. 8 and in the text. As expected, anti-IL5 antibodies reduced BM accumulation of activated eosinophils (Fig. 8c and d), thus confirming data obtained in the *de novo* Vk*MYC MM model. However anti-IL5 antibodies neither affected BM accrual of Th17 cells (Fig. 5b) nor

impacted disease development (Fig. 5a).

All together, these data support the concept that especially in the early phases of MM development and progression Th17 and eosinophils exert non-redundant and synergistic activities in supporting neoplastic plasma cell growth. Our results lend weight to the hypothesis that IL-17 acts both on neoplastic plasma cells and eosinophils, favoring the production by both populations of cytokines supporting plasma cell survival and proliferation. IL-6 should also favor the acquisition/maintenance of a Th17 skew (Fig. 7).

To better clarify this issue, we have modified the Discussion as follows: (page 19): *“The pro-tumor activity of IL-17 is not limited to its direct effect on neoplastic plasma cells expressing the IL-17R, but also through the local activation of eosinophils. Indeed, eosinophils were induced to produce TNF- α , IL-6 and likely others tumor-promoting cytokines upon stimulation with IL-17. IL-6 has long been known as a proliferative factor for MM cells. While neoplastic plasma cells can produce IL-6, the most accepted view is that the major source of this cytokine in the BM environment are BM stromal cells, osteoclasts and myeloid precursors cells from the early myeloblast to the intermediate myelocyte maturation stages⁵⁵. The latter population may contain eosinophils, which have been recently reported to support the early growth of murine neoplastic plasma cells in their BM niche⁴¹. Our data lend further credit to the role of eosinophils as key cells in the early neoplastic plasma cell niche, but do not exclude the role of additional IL-6-producing cells in the BM environment. The role of eosinophils in early MM is further supported by the finding that progression to MM was delayed in early-MM Vk*MYC mice only if treated with the combination of antibodies specific for IL-17, IL-17RA and IL-5, and therapeutic efficacy correlated with a reduced BM accrual of both Th17 cells and eosinophils. In more advanced MM, as the one reproduced in t-Vk*MYC MM mice, anti-IL5 antibodies are ineffective, and blocking the IL-17 signaling is sufficient to delay MM. Thus, our findings confirm that eosinophils are required for the maintenance of neoplastic plasma cells in the BM niche⁴¹ at the early stage of disease, and add the notion that IL-17 is one critical cytokine in the BM microenvironment that activates eosinophils to release factors supporting neoplastic plasma cells. As the role of IL-5 as growth factor for myeloma plasma cells is debated^{41,56}, and IL-5 should not impact on BM stromal cells, one mechanism by which anti-IL17, anti-IL-17RA, and anti-IL5 antibodies acted in Vk*MYC mice is a reduced accrual and survival of eosinophils and consequently of Th17 cells. While our data have highlighted a relevant crosstalk between eosinophils and Th17 cells in the BM of Vk*MYC mice, other cells within the tumor microenvironment produce IL-17, and also stromal cells respond to IL-17 by producing IL-6³⁰. Thus, our therapeutic approach should also target these cells”.*

Methodologies and statistics. Experimental details are missing for a number of experiments and some experiments are performed only once and contain insufficient amount of animals to make robust conclusions. Some examples are listed below.

As detailed below, we have added the requested information and repeated several experiments to make our conclusions more robust.

Specific Points

Microbiota:

How were mice from the two facilities matched in experiments (age, sex?). It is crucial that microbiota transfer experiments are performed, so littermate mice that differ only in their microbiota are compared. Were the experiments on IL-17KO mice performed with littermate controls? Where were these animals (and the controls) housed? Because of the possible effects of microbiota it is crucial that genetic experiments are performed with littermate controls.

We apologize for having been not clear enough. Details on age- and sex- matching have been added. All experiments were conducted with sex- and-age matched mice, and when specified mice

were littermates. Microbiota transfer experiments have been conducted and reported in Fig. 1e. We refer Reviewer #2 to the reply to her/his Major Points for details.

IL-17⁺ cell increase in conventional facility and sick mice (e.g. Fig 1G, 1E). This needs to be documented better. Frequency of total live cells is reported, which makes it unclear of whether this is due to increase in IL-17⁺ cells or decrease in another cell subset in the PP/BM. Total numbers or normalization to a reference population should be provided. What is the nature of these cells? Are they Th17 cells? If yes, frequencies within CD4 T cells can be presented.

We have added the absolute number of IL-17 cells in Fig. 1. To make room for new results from specific bacteria colonization, data with frequencies of IL-17⁺ cells in the Peyer's Patches and BM have been transferred to Suppl. Fig. 2. The nature of these cells is specified in more details in Fig. 2 and Suppl. Fig. 4.

The Kaede experiments mainly follow Th17 cells. Is this the main proposed IL-17 producing subset in the BM?

As reported in the text and in Suppl. Fig 4, Th-17 cells were the most represented IL-17⁺ cells in the BM of Vk*MYC mice. More in details: (page 11) “*Several immune cells produce IL-17⁵. Indeed, the BM of Vk*MYC mice contained measurable populations of CD3⁺CD4⁺ (Suppl. Fig. 4a), CD11b⁺Gr1⁺ (Suppl. Fig. 4b), Nk1.1⁺ CD90.2⁻ (Suppl. Fig. 4c) and Lin⁻CD90⁺CD127⁺ cells producing IL-17 (Suppl. Fig. 4d), of which T helper type 17 (Th17) cells were the most represented (Suppl. Fig. 4e)*”.

Is colonization with Th17 cell-inducing bacteria, e.g. SFB or Prevotella, sufficient to promote disease?

This appears to be the case. These new data are reported in Fig. 1e and in the text as detailed in the Major Points section of this reply.

Fig 1N – it seems that there is a difference between IL17KO and IL-17KO+Abx at the first timepoint. This experiment is important to show dependence on IL-17, but only 5 mice/group in a single experiment are presented and statistics are missing. WT+ABX condition is also missing. More experiments/animals are needed. Similarly on Fig 1O the animal numbers are low in a single experiment. Moreover, it seems that the penetrance of disease is low in the controls (3 out of 5 mice do not show disease). Why is that? Also, can the authors clarify what is plotted on this figure? What is % of M-spike?

In the revised version of the paper we have now included additional experiments increasing the total number of mice and including the control group of IL17^{WT} mice treated with ABX. Statistical analyses have been added. Penetrance of the disease in untreated animals varies in the 80-100% range, as expected (Chesi M. et al. Blood 2012). To clarify how M-spike was calculated we have modified the in the legends and Materials and Methods section (e.g., pag. 21: “**Serum Protein Electrophoresis.** Mouse blood was periodically collected in Eppendorf by retro-orbital sampling. Semi-automated electrophoresis was performed on the Hydrasys instrument (Sebia, Lissex, France). According to the manufacturer's instructions, 10 μ L of undiluted serum were manually applied to the Hydragel agarose gels (Sebia). The subsequent steps: electrophoresis (pH 9.2, 20W constant current at 20°C), drying, amidoblack staining, de-staining and final drying were carried out automatically. The use of Hydrasys densitometer and Phoresis software (Sebia) for scanning resulting profiles provided accurate relative concentrations (percentage) of individual protein zones. M-spike levels were calculated as total gamma globulins/albumin ratio (G/A)¹⁷”).

Abx treatment – when do M-spikes appear in treated mice? How delayed is the appearance of M-spike and what is the difference in quantification of M-spike? Kinetics similar to the ones in Fig 1A/2A and 2B can be presented.

We apologize for the lack of clarity regarding this experiment. As suggested by Reviewer #2, we analyzed and reported the data as incidence of M-spike.

Fig 1E – the vehicle graph is incomplete. Do any of the animals survived beyond 45 days?

As reported in Chesi M. et al. (Blood 2012) disease may not appear in 10-20% of the mice in the t-Vk*MYC MM model. This has been specified as follows: (page 7) “*As expected*¹⁷, *three weeks after transplantation the paraprotein was measurable in sera of 80% of untreated mice, but none of the mice treated with antibiotics showed signs of disease (Fig. 1d and Suppl. Fig. 1b). This did not appear to be due to a direct effect of the antibiotics on plasma cell survival, because the M-spike appeared later on in several antibiotic-treated mice (Fig. 1d). Importantly, at the time that all the untreated t-Vk*MYC MM mice with M-spike succumbed of the disease, all antibiotic-treated mice were still alive, and overall survival was improved in the latter group (Suppl. Fig. 1b)*”.

Legend for Figure 2D-G is wrong

We apologize for the mistake. Figure legends have been revised and corrected.

Suppl Table 1 is missing

We apologize for having not uploaded the Table, which is now available in the Supplementary Data section.

Figure 4C is impossible to understand. It should be removed or replaced with a more relevant analysis.

We agree with both reviewers that the heat maps were not clear enough to be shown in the figures of the manuscript. This having been said, we are still convinced they contained relevant information for the comprehension of the work. Thus, we modified and included them in the supplementary material (Suppl. Figs. 5 and 6).

For some figures it will be helpful to include representative FACS plots in order to understand better the proposed phenotypes/changes. For example on Figure 5 where IL-6 production is measured, especially because MFI changes are used. MFI changes could be supported by a more reliable method for protein quantification, e.g. ELISA

As requested, we added a representative dot plot in Fig. 5. Because eosinophils were cultured with BM sera, which already contain IL-6, we thought intracellular staining was more precise and selective.

Fig 5F – this figure is difficult to read. Maybe some statistics and quantification data for MCP-3 expression under different treatment groups, rather than a heat map, would be better.

As commented above, the heat map previously reported in Fig. 5 has been modified and transferred in the Supplementary Data file as Suppl. Fig. 6. Quantification of MCP3 in the different experimental conditions is now reported in the text at pag. 13: “*Consistently, we detected a trend toward higher levels of MCP-3, which is known to attract and activate eosinophils*³⁷ *in the BM of Early-MM mice when compared to Late-MM [Early-MM 376.9 ±128.5 pg/ml (mean±SE; n=7); Late-MM 100.2 ± 15.45 pg/ml (n=5); WT 169.8 ± 46.7 (n=5); Suppl. Fig. 6]*”.

Missing or incomplete Methods:

We complied with all these criticisms, and modified the text accordingly.

Description of the two animal housing facilities is missing

We have added a detailed description of each animal facility (e.g., pag. 22: “*All these mice were maintained under specific pathogen-free (i.e. the rodents were housed in isolated rooms, fed sterilized food and water, and routinely tested and determined free of designated pathogens capable of interfering with research objectives; SPF) conditions in the San Raffaele facility and experiments were performed according to state guidelines and approved by the European Community Guidelines (Authorizations #574, #1147, #863)*”).

The nature of the controls used should be noted for all experiments. Are they littermates, age, sex-matched, etc.?

These details have been added.

What is meant by “Vk*MYC mice were backcrossed into IL-17KO mice for 6 generations”?

To avoid genetic drifting, Vk*MYC mice were backcrossed into IL-17^{KO} mice for at least 6 generations before generating homozygous Vk*MYC IL-17^{KO} breeding pairs. This has been specified in the text (page 20).

Are Vk*MYC, IL-17KO and Kaede mice on B6 background?

Kaede-transgenic mice were on a C57BL/6 background, as now specified in the text at page 21: “*Kaede-transgenic mice on a C57BL/6 background were generated by Dr. Miwa Yoshihiro at the University of Tsukuba, Japan* ²⁴”.

M-spike quantification method is not described.

M-spike quantification method is now specified at page 22 as reported above.

How was early and late stage defined on Fig 2?

This is specified in ref. 31 and in the text at page 10: “*Additionally, disease progression [i.e., M-spike $\geq 6\%$, which is characteristic of symptomatic, Late-MM; ref. ³¹] was delayed in Vk*MYC IL-17^{KO} mice (Fig. 2b) when compared to fully immunocompetent Vk*MYC mice, thus demonstrating that IL-17 is also a precocious propeller of MM in this model. As expected, WT mice never progressed to MM (Fig. 2b).*

*As our results suggested that IL-17 is involved in early phases of disease (Fig. 2a), we quantified IL-17⁺ cells (Fig. 2c) in the BM of both asymptomatic (Early)- and symptomatic Late-MM Vk*MYC mice ³¹”.*

How were PP single cell suspensions prepared?

The method is now reported at page 25: “*Peyer’s Patches were removed from the Small Intestine, and gently disaggregated with the help of tweezers*”.

In vitro polarization – more detail is needed. What is BM serum and how was it prepared?

To clarify these issues we have modified the text as follows: pag. 26: “*Th17 polarization in vitro. OTII splenocytes were cultured in complete IMDM for 7 days under stimulation with anti-CD3/CD28 Dynabeads (4×10^5 beads/ 2×10^5 cells; Invitrogen, Thermo Fisher, Milan, IT), and in the presence of either the combination of IL-6 (20ng/ml, PeproTech, tebu-bio, Milan, Italy), TGF- β 1 (2ng/ml, R&D Systems, Minneapolis, MN), anti-IL-4 (10ug/ml, BD Biosciences) and anti-IFN- γ antibodies (10ug/ml, Biologend). Alternatively, stimulated cells were cultured in the presence of BM serum (1:25 final dilution). After 7 days OTII cells were tested for intracellular cytokine production by flow cytometry*”; and pag. 25 “*Collection of BM serum and cells. Each femur devoid of epiphyses was placed into a 0.5 ml eppendorf tube whose bottom was pierced with a 16G needle. The pierced eppendorf tube containing the bone was subsequently placed into a 1.5 ml eppendorf tube and centrifuged (HeraeusTM PicoTM 17 Microcentrifuge, ThermoFisher Scientific, Waltham, MA USA)*

for few seconds. The BM pelleted material, containing both serum (approximately 10 μ l) and cells was resuspended in 100 μ l PBS, and used for flow cytometry analyses. Alternatively, the resuspended material was centrifuged again to separate diluted serum from cells, and stored at -80°C”.

Reviewers' comments:

Reviewer #1 (Remarks to the Author):

In the revised version, the authors adequately respond to the points raised by me.

Reviewer #2 (Remarks to the Author):

The manuscript is significantly improved, especially in regards to characterizing the role of IL-17 and eosinophils. The authors have also provided new data that colonization with *Prevotella* can increase disease. However, definitive role for microbiota remains to be demonstrated and this is one of the main conclusions of the manuscript. As noted in the previous review, the results with mice from different facilities are interesting and suggestive of microbiota effects, but are far from conclusive. From the new clarification provided by the authors, it seems that the mice are not only in different facilities and institutions, but on different continents. Therefore, multiple other environmental factors could be in play. Animals from the same facility, but colonized with different microbiotas need to be directly compared to claim microbiota effects. The easiest experiments would be to colonize antibiotic treated mice with microbiotas from different facilities and compare disease/phenotypes, or transfer microbiota from CNV mice to SPF mice and show transfer of phenotype. The *Prevotella* experiment is missing important controls and therefore only adds additional variables. Also, as is, this experiment cannot faithfully explain the difference in phenotypes between the two facilities (see below).

Other points:

FMT from CNV microbiota or microbiota isolates into SPF mice need to be performed to claim microbiota effects.

The terms CNV and SPF mice for the different animal sources are inaccurate. All of the mice are conventionally-raised and specific-pathogen free in the sense used in the manuscript. It is better to use Facility1 vs Facility2 or US vs Italy designations that better reflect the actual source of the mice.

The *Prevotella* experiment is interesting, however, results need to be compared to colonization with other bacteria. Additional detail on the *Prevotella heparinolytica* is warranted? Is this particular species present in the normal gut microbiota? Is it present in the CNV mice? Does P.h. behave as commensal in this experiment? What are the colonization levels in feces? Is there systemic infection that may be inducing inflammatory cytokines, including IL-17 that may be exacerbating disease? What is plotted on X axis on Fig 1e - "days since disease appearance"? Is there a difference in disease onset?

Figure S5 - Min/Max should be replaced with actual values

We thank all referees for their constructive comments that have helped to further improve the quality of our manuscript. We have reorganized our manuscript on the basis of a number of new critical findings. Furthermore, we have addressed in full the critiques and constructive suggestions of Referee #2, which have also helped in strengthening and corroborating our original conclusions, specifically, the relevance of the microbiota and Th17 cells in MM progression. We believe that this work is relevant at a number of levels and might strongly impact on the current paradigm underlying the role of the microbiota in regulating cancer progression. Taken together, our findings lead to a number of critically important conclusions:

1. We demonstrate that the microbiota composition speeds multiple myeloma progression;
2. we also show that Th17 cell-eosinophil axis promotes neoplastic plasma cell survival;
3. finally, we collected data to show that IL17 and IL5 blocking antibodies limit multiple myeloma progression *in vivo*. Given that both anti-IL5 and anti-IL17 blocking antibodies are currently in the clinic for the treatment of immune-mediated diseases and are well tolerated, our findings might have rapid clinical applications.

Referee #2:

The manuscript is significantly improved, especially in regards to characterizing the role of IL-17 and eosinophils. The authors have also provided new data that colonization with *Prevotella* can increase disease. However, definitive role for microbiota remains to be demonstrated and this is one of the main conclusions of the manuscript.

As noted in the previous review, the results with mice from different facilities are interesting and suggestive of microbiota effects, but are far from conclusive. From the new clarification provided by the authors, it seems that the mice are not only in different facilities and institutions, but on different continents. Therefore, multiple other environmental factors could be in play. Animals from the same facility, but colonized with different microbiotas need to be directly compared to claim microbiota effects. The easiest experiments would be to colonize antibiotic treated mice with microbiotas from different facilities and compare disease/phenotypes, or transfer microbiota from CNV mice to SPF mice and show transfer of phenotype. The *Prevotella* experiment is missing important controls and therefore only adds additional variables. Also, as is, this experiment cannot faithfully explain the difference in phenotypes between the two facilities (see below).

We thank the reviewer for these important suggestions that have been fully addressed. The new results have been incorporated in the revised version of the manuscript.

1. FMT from CNV microbiota or microbiota isolates into SPF mice need to be performed to claim microbiota effects.

As correctly summarized by Referee #2, our analyses were performed comparing animals housed in two different facilities, located in Italy and USA. The experiments and analyses of CNV mice (now renamed US1 mice) are dating before 2014. At that time, breeding of the Vk*MYC colony in US1 was made by crossing genetically modified mice with wild type (WT) littermates. Later on, for colony maintenance reasons and to avoid genetic drifting, Vk*MYC male mice were bred with C57BL/6J females purchased from the Jackson lab. The microbiota imported from the purchased mice substantially modified the microbiota composition of Vk*MYC mice, which we named US2. Indeed, unweighted UniFrac principal component analyses (β -diversity) of stools from US1 and US2 (recently collected from mice derived from breeding placed after 2017) showed a clear segregation between the US1 and US2 or SPF (from now renamed IT) cohorts of mice, and showed large differences in bacterial species between US1 and US2 mice, irrespective of being housed in the same facility and institution. Unexpectedly, we found that stools from US2 mice were not enriched anymore of *Prevotellaceae* species. We have now included these data in Figure 2b and c of the revised manuscript. Accordingly, we found that the survival curve of IT mice challenged with Vk12598 cells and subjected to FMT with stools from US2 mice was paralleling the curve obtained from mice that have received IT stools. For Referee #2 perusal, we reported these data in Annex I at the end of this rebuttal.

These unexpected findings ended up as being very informative. Indeed, when we compared the incidence of M-spike in US2 Vk*MYC mice, which do not contain enrichment of *Prevotellaceae* in their stools, with *Prevotellaceae*-rich US1 Vk*MYC mice from the same animal facility, we found that the M-spike was readily detectable by 20 weeks of age in the blood of about 30% US1 Vk*MYC mice, whereas age- and sex-matched US2 Vk*MYC mice did not show signs of disease for additional 10 weeks, time at which more than 60% of the US1 Vk*MYC mice had a detectable M-spike. Thus, myeloma appearance in US2 Vk*MYC mice paralleled *Prevotellaceae*-poor IT Vk*MYC mice. These findings strongly support the role of *Prevotellaceae* in favoring myeloma aggressiveness, and confirm what we have found by infecting mice with *Prevotella heparinolytica*. We have incorporated the new findings in Figure 1a of the revised manuscript.

2. The terms CNV and SPF mice for the different animal sources are inaccurate. All of the

mice are conventionally-raised and specific-pathogen free in the sense used in the manuscript. It is better to use Facility1 vs Facility2 or US vs Italy designations that better reflect the actual source of the mice.

As correctly suggested by the referee, we have re-named mice belonging to the different colonies.

The Prevotella experiment is interesting, however, results need to be compared to colonization with other bacteria. Additional detail on the Prevotella heparinolytica is warranted? Is this particular species present in the normal gut microbiota? Is it present in the CNV mice? Does P.h. behave as commensal in this experiment? What are the colonization levels in feces? Is there systemic infection that may be inducing inflammatory cytokines, including IL-17 that may be exacerbating disease? What is plotted on X axis on Fig 1e - “days since disease appearance”? Is there a difference in disease onset?

We thank Referee #2 for asking if other bacteria were accelerating myeloma. Indeed, the new experiments substantially increase the strength of our findings. To comply with the referee's request, Vk12598-challenged and antibiotic-treated mice were infected by gavage either with *P. heparinolytica* or *P. melaninogenica*. The latter was chosen because it has been implicated in rheumatoid arthritis (ref 48 in the manuscript). Neither groups of mice lost weight nor showed signs of systemic infection at gross pathology. We have found that only *P. heparinolytica* accelerated disease and associated with IL-17⁺ cell enrichment in the gut and the bone marrow of the treated mice, as now reported in Figs. 1e, h and j.

Prevotella was measured in the stools of infected mice by RT-PCR, as it is now specified in the Materials and Methods section of the manuscript. *P. melaninogenica* 2^{-ΔCT}: 129605 ± 83210 and *P. heparinolytica* 2^{-ΔCT}: 322139 ± 122909.

Referee #2 also raised important issues related to *Prevotellaceae*. *Prevotellaceae* are normal component of the microbioma both in humans and in mice, and are considered commensals. Only a few strains were reported to behave as pathobionts and cause opportunistic infections (ref. 24 of the text). As detailed in Figure 1c, *Prevotellaceae* were detected in the stools of all mice, but in different amounts. To our knowledge, *Prevotellaceae* are symbionts also in C57BL/6 mice. Indeed, we looked for signs of gut inflammation in our mice, and we never detected histopathologic signs of colitis in mice housed either in Italy or in the USA. This was already reported in the previous versions of the manuscript [“Accordingly, at pathologic examination, we did not find relevant signs of inflammation in the gut of CNV animals (data not shown).”]. In the first revision of the

manuscript we have also added the following sentence: (page 16) “*Thus, Prevotella species are primary suspects also in humans, in which the increased abundance of these bacteria at mucosal sites has been associated with Th17-mediated diseases including periodontitis and rheumatoid arthritis.*” Thus, both in humans and in the Vk*MYC model *Prevotellaceae* favour rather than cause disease.

The new Fig. 1e reports the survival of Vk12598-challenged and antibiotic-treated mice that were sham-gavaged or infected with either one of the two *Prevotella* strains. To mimic what we think happens in humans, mice were repeatedly exposed to *Prevotella* during both disease development and progression. As our aim was to demonstrate that *P heparinolytica* accelerates disease progression, and M-spike appearance is variable in these mice, mouse survival was plotted since the day we documented the presence of the M-spike. To make it more explicit, we modified the legend as follows: “Days since M-spike appearance”.

4. Figure S5 - Min/Max should be replaced with actual values

We have modified the figure as requested by Referee #2.

Calcinotto et al. Annex 1

Figure Annex 1. Overall survival (Kaplan-Meier plot) of t-Vk*MYC MM mice treated with vehicle by gavage (Vehicle), or subjected to fecal microbiota transplantation with stool preparations either from IT (FMT IT) or US2 (FMT US2) mice. Long-rank (Mantel-Cox) test. Exact P values and the number of biologically distinct mice is shown.

Method

FMT experiment. Ampicillin (1g/L; Sigma-Aldrich), Vancomycin (0,5 g/L; Sigma-Aldrich), and Neomycin (1g/L; Sigma-Aldrich) were added to the drinking water of 6 week-old WT C57BL/6J, and Metronidazole (2mg/mouse; Sigma-Aldrich) was administered by oral gavage 3 times per week. Two weeks later, antibiotic-treated animals received FMT of feces from C57BL6 mice housed in IT AF or US2 AF for 3 consecutive days/week, until the end of the experiment. Feces were collected from at least 8 different donors and immediately re-suspended in PBS at 50mg/ml. Each recipient mouse received an oral gavage of 200 μ l. After two weeks of FMT, mice were challenged I.V. with 1×10^6 Vk12598 cells. For disease monitoring, mouse blood was collected by retro-orbital sampling once a week starting from the third week since tumor challenge, and analyzed by Serum Protein Electrophoresis as described above.

REVIEWERS' COMMENTS:

Reviewer #2 (Remarks to the Author):

The new data with *P. heparinolytica* and *P. melaninogenica* are a significant improvement showing that one microbe promotes disease, while a closely-related one does not. This indeed, shows that *P.h.* can accelerate disease in a specific manner, which is important. Unfortunately, the experiments on Figure 1 suffer from the same drawbacks as before, and are not conclusive to claim that the observed differences are due to differences in the microbiota. FMT transplantation is needed for this to be proven. Indeed, the authors performed FMT, but the results of these experiments are negative (the FMT did not change the disease progression). This reviewer appreciates that there is a difference in the microbiota between the original US1 and the US2 mice used for the FMT, but in the absence of direct evidence, whether the difference in the phenotype between US1 and IT/US2 is due to these changes in microbiota remains speculative.

Other points:

It seems that all of the experiments with US1 mice were performed before 2014 and all of the experiments performed with US2 and IT mice, after 2015. This means that the comparisons on Figure 1 are comparisons between different experiments. How and when the experiments were performed needs to be clearly explained in the text and the figure legends.

In the results it is stated that "To identify constituents of the microbiota, fecal samples simultaneously collected from mice housed in the different animal facilities were subjected to 16S rDNA-based amplicon sequencing." Considering that the samples were collected in different years, it is not clear what is meant by "simultaneously collected".

We thank Referee #2 for the additional comments to our manuscript that further improved it and balanced our conclusions.

The new data with *P. heparinolytica* and *P. melaninogenica* are a significant improvement showing that one microbe promotes disease, while a closely-related one does not. This indeed, shows that *P.h.* can accelerate disease in a specific manner, which is important.

We thank the reviewer for her/his very encouraging comment. Indeed, having identified a strain of *Prevotella* involved in murine MM progression might have relevant implications also in the human disease. These data also support a potential role of the gut microbiota in extramucosal tumors.

Unfortunately, the experiments on Figure 1 suffer from the same drawbacks as before, and are not conclusive to claim that the observed differences are due to differences in the microbiota. FMT transplantation is needed for this to be proven. Indeed, the authors performed FMT, but the results of these experiments are negative (the FMT did not change the disease progression). This reviewer appreciates that there is a difference in the microbiota between the original US1 and the US2 mice used for the FMT, but in the absence of direct evidence, whether the difference in the phenotype between US1 and IT/US2 is due to these changes in microbiota remains speculative.

We agree with Referee #2 that FMT is the ideal experiment to demonstrate a direct relationship between gut microbiota and MM progression. We respectfully disagree with the conclusion that the results we obtained with FMT were negative. In our opinion those experiments were highly informative and, together with the evidence that *Prevotellaceae* were substantially reduced in US2 stools strongly support the role of *P. heparinolytica* in MM progression in the mouse model used. Being *P. heparinolytica* a commensal bacterium, our data strongly support a potential role of the gut microbiota in MM progression.

To comply with this criticism, and in agreement with the Editor's suggestion, we toned down our conclusions about the microbioma, and we highlighted plausible caveats in our experimental design. Modifications can be easily found highlighted in yellow in the abstract and the text of the manuscript.

Other points:

It seems that all of the experiments with US1 mice were performed before 2014 and all of the experiments performed with US2 and IT mice, after 2015. This means that the comparisons on Figure 1 are comparisons between different experiments. How and when the experiments were performed needs to be clearly explained in the text and the figure legends.

In the results it is stated that “To identify constituents of the microbiota, fecal samples simultaneously collected from mice housed in the different animal facilities were subjected to 16S rDNA-based amplicon sequencing.” Considering that the samples were collected in different years, it is not clear what is meant by “simultaneously collected”.

We better explained in the text and the Methods of the revised version of the manuscript how we collected stools from the mice. Due to space limitations, in the legend to Fig. 1 we refer to the Methods section for further details on how stools were collected. The modifications can be found highlighted in yellow in the revised version of the manuscript. It is not correct to write that stools from IT mice were collected after 2015. Indeed, stools were simultaneously collected in US1 and IT in 2013 and in US2 and IT in 2017. Data obtained from the two IT analyses were substantially superimposable.